# Beyond the Linear Separability Ceiling: Aligning Representations in VLMs

## Abstract

A challenge in advancing Visual-Language Models (VLMs) is determining whether their failures on abstract reasoning tasks, such as Bongard problems, stem from flawed perception or faulty top-down reasoning. To disentangle these factors, we introduce a diagnostic framework centered on the Linear Separability Ceiling (LSC), the performance achievable by a linear classifier on a VLM's raw visual embeddings. Applying this framework to state-of-the-art VLMs, we uncover a pervasive "alignment gap", where most models fail to generatively outperform the linear separability of their own representations. We find that the few models surpassing this ceiling do so via two mechanisms: by further refining visual representations into a more linearly separable format or by executing non-linear decision logic. We demonstrate that this bottleneck is not a fundamental limitation but a solvable visual alignment issue. By augmenting standard next-token prediction with a contrastive objective, our method restructures the visual manifold into a more one-dimensionally linear geometry, improving image-to-image comparison and enabling models to significantly surpass the LSC on abstract binary classification tasks.

## 1 Introduction

A challenge for state-of-the-art Visual-Language Models (VLMs) (Radford et al., 2021) is understanding the root of their frequent failures on abstract tasks. Is the bottleneck flawed bottom-up visual perception, which builds representations from sensory input (Marr, 2010), or flawed top-down reasoning that interprets that input using prior knowledge (Gregory, 1970)? This question highlights a persistent gap between machine and human cognition, particularly on visual puzzles (Wüst et al., 2024) and tasks VLMs solve in text but not in visual formats (Park et al., 2025). While previous studies have explored this perception-reasoning interface within modern VLMs for abstract tasks like Bongard problems, the primary cause of failure has been difficult to isolate, lacking an empirical method to quantify this gap (Vaishnav & Tammet, 2025; Małkiński et al., 2025).

To address this ambiguity, this paper introduces a diagnostic framework centered on the Linear Separability Ceiling (LSC), a measure of the performance a linear classifier can achieve on a VLM's raw visual embeddings. The LSC provides a baseline for the quality of the initial visual representations, establishing a benchmark that the model's end-to-end (non-linear) reasoning must surpass to demonstrate added value.

Our focus on computational non-linearity is complemented by recent work on representational geometry by Engels et al. (2025) that challenges the Linear Representation Hypothesis (LRH). They find that concepts like days of the week and months of the year form irreducible, circular features in activation space. This *representational* non-linearity implies a need for the *computational* non-linearity we investigate; as they show, models use these exact circular features to perform tasks involving modular arithmetic. Their work thus offers a concrete, mechanistic example of the structured features our hypothesized (non-linear) reasoning pathways may operate on.

Applying this framework reveals our central finding: an "alignment gap," a concept that extends known representational issues like the "modality gap" (Yaras et al., 2022) to the interface between perception and reasoning. We find that a model's reasoning pathways are often not aligned with its own high-quality visual representations. This causes the generative performance of most leading VLMs to be statistically no better than the LSC on their own visual embeddings.

The models that surpass this ceiling do so via two strategies: by further refining representations into a more linearly separable format, effectively extending the bottom-up perception process, or by executing a non-linear decision logic that functions as a form of top-down reasoning. Strong evidence for this second pathway comes from the success of postfix tuning (Li & Liang, 2021; Lester et al., 2021), a method which by design cannot alter the initial representations.

## 2 RELATED WORK

**Abstract visual reasoning in VLMs.** The evaluation of VLMs has evolved from foundational benchmarks for single-image understanding, such as captioning (Lin et al., 2014) and VQA (Goyal et al., 2019), to more rigorous tests of reasoning. The complexity of these evaluations has increased along two axes. First, at the vision-language interface, benchmarks now demand deeper reasoning. This includes VQA variants that test for compositional structure and spatial skills like GQA (Hudson & Manning, 2019), or probe commonsense understanding as in VCR (Zellers et al., 2019). Complementing these, datasets like Winoground (Thrush et al., 2022) specifically isolate visio-linguistic compositional abilities. Second, in the purely visual domain, a distinct set of benchmarks evaluates abstract reasoning by removing the linguistic component entirely. This category includes Raven's Progressive Matrices (RPMs) (Zhang et al., 2019), the Abstract Reasoning Corpus (ARC) (Chollet, 2019), and Bongard tasks (Bongard, 1970), which are the focus of this work.

**VLM architectures and modality fusion.** A key architectural differentiator in VLMs is the strategy for fusing modalities (Gadzicki et al., 2020; Shukor et al., 2025). While **early-fusion** models like Chameleon (Team, 2024) create a joint representation from raw inputs, the now-dominant **late-fusion** approach first processes images with a dedicated vision encoder. In its most common form, exemplified by LLaVA (Liu et al., 2023), the resulting visual embeddings are mapped into the language model's space via a simple projection layer. A more deeply integrated variant is cross-attention fusion, pioneered by Flamingo (Alayrac et al., 2022) and BLIP (Li et al., 2022), which insert cross-attention layers within the LLM (Lin et al., 2021). Our framework is designed for late-fusion models where initial visual representation can be isolated.

**Linear Representation Hypothesis (LRH)** posits that concepts within neural networks are represented by linear structures in activation space. This is commonly observed as compositional linearity, where the embeddings of composite concepts can be decomposed into the sum of their constituent parts (Trager et al., 2023), and appears fundamental to transformer decoders, whose sequential layer transformations are often nearly linear (Razzhigaev et al., 2024). Theoretical work supports these empirical findings, suggesting that the next-token prediction objective inherently biases models towards learning representations that are linear transformations of latent concepts (Liu et al., 2025). This hypothesis suggests that the learned representations should be linearly separable — that is, distinguishable by a simple linear classifier. This property is critical for generalization, enabling **linear transferability** where a classifier trained on a source domain remains effective on a new, related target domain (HaoChen et al., 2022). The emergence of such a well-ordered geometric space is not a coincidence but a predictable consequence of high-dimensional geometry. Stochastic separation theorems establish that in high dimensions, any given point in a random set can be separated from the others by a hyperplane with high probability, even if the number of points in the set grows exponentially with the dimension (Sidorov & Zolotykh, 2020).

**Representation alignment and the modality gap.** Research into representation alignment has identified critical bottlenecks in VLMs, which can occur either in the initial visual encoding or in the subsequent reasoning process (Chia et al., 2024; Zhang et al., 2024). A key example of such an alignment challenge is the modality gap: a widely observed phenomenon where the learned representations for different modalities exhibit distinct structures or occupy separate regions within the joint embedding space, even when they are semantically aligned (Yaras et al., 2022). This systematic separation, particularly noted in contrastively trained models, can hinder cross-modal alignment and the seamless integration of information (Qiu et al., 2024), potentially impacting the discriminative power of these representations. Our framework allows us to quantitatively assess the impact of such issues by establishing the linear separability ceiling as baseline. This concept of a representational performance ceiling is not new; for instance, prior work has identified a "contrastive learning ceiling" in semantic textual similarity (Zhang & Li, 2024).

**Training and adapting.** To bridge the modality gap and enhance reasoning, various adaptation strategies have been developed. While some work proposes novel architectural modifications (Bigverdi et al., 2025; Kolner et al., 2025) or reinforcement learning techniques (Li et al., 2025; Huang et al., 2025), our work focuses on training objectives. Vision encoders are typically pre-trained with objectives like contrastive learning, as seen in CLIP (Radford et al., 2021), or combined contrastive-generative objectives, as in CoCa (Yu et al., 2022). Building on this, some methods augment the standard next-token prediction objective with an auxiliary contrastive loss to explicitly align representations (Ouali et al., 2025; Wu et al., 2025; Ak et al., 2024). While these approaches typically tackle the modality gap by aligning image-to-text pairs, we align image-to-image pairs. To apply such strategies efficiently, parameter-efficient fine-tuning (PEFT) methods like Low-Rank Adaptation (LoRA) (Hu et al., 2022) and prompt tuning (Li & Liang, 2021; Zhou et al., 2022) are widely adopted (Lester et al., 2021; Jahan et al., 2025). We investigate whether strategies that co-optimize for enhanced representational discriminability alongside generative accuracy enable VLMs to better leverage their internal visual representations.

## 3 EXPERIMENTAL SETUP

Our analysis centers on abstract visual reasoning using Bongard-style tasks, which require a model to infer a rule from positive and negative examples to classify a query image. A representative example of this task structure is shown in Figure 1.

Figure 1: An example of the Bongard HOI task. The model must infer a common rule, in this case, "a person is performing a jump on a motorcycle," from the positive examples and determine if the query image follows this rule, which the negative examples do not.

**Datasets.** We use two Bongard-style datasets: Bongard OpenWorld (Wu et al., 2024), split into train (500), validation (100), and test (500) samples with distinct semantic components across splits; and Bongard HOI (Jiang et al., 2022). For HOI, we use its original splits, with a 4000-sample balanced training set. Its validation ($4 \cdot 100$) and test ($4 \cdot 200$) sets are categorized by concept novelty (seen/unseen object/action). Each sample contains 6 positive, 6 negative, and 1 query image. For cross-domain generalization, we also evaluate on the Winoground benchmark, a text-image retrieval task designed for testing compositional reasoning.

**Models.** Baseline performance was established on multiple SoTA VLMs: Phi 3.5 vision 4.2B (Microsoft, 2024), Pixtral 12B (MistralAI, 2024), Gemma3 4B and 27B (Google, 2025), InternVL3 14B (Chen et al., 2024), and Qwen 2.5 VL 7B and 72B (Alibaba, 2025). Subsequently, we applied PEFT to Gemma3 4B, Phi and Pixtral.

**Evaluation.** For all fine-tuning experiments, we saved model checkpoints every epoch and selected the one with the best performance on the validation set. Baselines and best-performing checkpoints were evaluated on their respective test splits. Models demonstrate consistent performance across all conceptual novelty splits (see Appendix L), so we report the average for brevity.

# 4 A FRAMEWORK FOR DECOMPOSING VLM REASONING

To diagnose whether VLM failures on abstract reasoning tasks stem from flawed perception or reasoning, we introduce a diagnostic framework. Centered on a non-parametric linear probe, our framework assesses the discriminability of visual embeddings, independent of the generative process, to disentangle representation quality from LLM processing effectiveness. We benchmark performance across various prompt structures, including interleaved vs. labeled image presentation and direct vs. Chain-of-Thought (CoT) prompting (see Appendix C).

## 4.1 METHODOLOGY

We then extract multi-token image embeddings at two key stages: the **vision** stage, which captures initial representations from the vision encoder, and the **final** stage, which captures contextualized representations from the LLM's final hidden state (detailed in Appendix D). We aggregate these sequences into single vectors, $\vec{v}_i$, using mean pooling and L2 normalization. This is a standard technique for producing a single embedding from a sequence of tokens (Reimers & Gurevych, 2019), that preserves angular properties relevant to cosine similarity. To quantify the discriminative quality of these embeddings, we probe them using a nearest-centroid classifier. We selected this nearest-centroid strategy as it proved more performant than a nearest-neighbor approach as it mitigates the influence of outlier examples and captures a more robust prototype of the abstract concept (Hastie et al., 2009), as shown in Appendix E. We then compute prototype vectors for the positive ($P$) and negative ($N$) example sets by averaging their respective image vectors (e.g., $\vec{c}_P = \mathrm{mean}(\vec{v}_{p_1}, \ldots, \vec{v}_{p_k})$). A query image ($Q$) is then classified based on its cosine similarity to these centroids (i.e., comparing $\cos(\vec{v}_Q, \vec{c}_P)$ to $\cos(\vec{v}_Q, \vec{c}_N)$).

## 4.2 DIAGNOSTIC METRICS AND CLASSIFICATION

Our framework dissects VLM performance by statistically comparing three key accuracies: the end-to-end generative accuracy, the linear probe accuracy on initial *vision* embeddings, and on *final* contextualized embeddings. Grounded in the linear representation hypothesis, which posits that concepts are encoded as linear structures within a model's activation space, we define linear probe accuracy on initial vision embeddings as the **linear separability ceiling (LSC)**, quantifying the baseline discriminative power of the visual representations, a primary goal of contrastive pre-training. To classify a model's performance, we perform a statistical significance test on the difference between its generative accuracy and its LSC, process visualized in Figure 2.

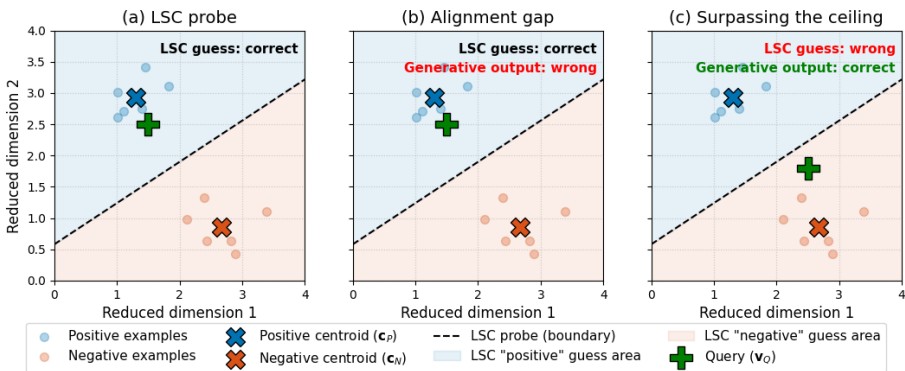

Figure 2: Visual examples of the diagnostic framework and its classifications. The LSC probe (dashed line) is the linear boundary between positive ($\mathbf{c}_P$) and negative ($\mathbf{c}_N$) centroids. **(a) LSC probe:** the query ($\mathbf{v}_Q$) is correctly classified by the linear probe. **(b) Alignment gap:** classified when generative accuracy is on average not statistically superior to the LSC ($p \geq 0.05$). The LSC probe succeeds, but the generative model fails. A non-linear model is expected to outperform its linear probe; failure to do so suggests a misalignment. **(c) Surpassing the ceiling:** classified when generative accuracy is on average statistically superior to the LSC ($p < 0.05$). The LSC probe fails, but the generative model succeeds. For illustration, assume reduced dimensions maximally preserve linear properties.

This approach accounts for the uncertainty in both measurements by analyzing the standard error of their difference (Burns & Dobson, 1981). A Chi-squared ($\chi^2$) test is then used to assess the statistical dependence between paired, trial-by-trial correctness of the model's generative predictions and the predictions from our linear probes ($p < 0.05$).

## 5 FINDING: A PERVASIVE ALIGNMENT GAP

Our framework, applied to 8 SoTA VLMs across two datasets and tested with 8 prompt formats each, reveals a pervasive **alignment gap**. As shown in Figure 3, a model's generative performance is highly variable and frequently fails to surpass its own LSC. We identify this widespread issue as an alignment gap, where a model's reasoning pathways are not effectively aligned with its own visual representations. Full results are in Appendix F.

Figure 3: Vertical axis is the generative performance across different prompts, horizontal axis is the linear probe classification accuracy. The datapoints in the green region are the instances when the model is successfully using its reasoning pathways to generatively classify the query image significantly better than a linear probe on inputs would, surpassing the linear separability ceiling. In the red region, however, alignment gap persists and the non-linear nature of VLM is failing to outperform a linear classifier. Statistical comparison accounts for 95% confidence intervals of both metrics.

### 5.1 ANALYSIS OF LSC-SURPASSING REASONING PATHWAYS

While most configurations are constrained by this bottleneck, a few successful models reveal how it can be overcome. Our analysis shows that successful models channel the transformer's iterative refinement process into two distinct computational strategies. These reasoning pathways are distinguished by the geometric transformations applied to the visual embeddings, with Table 1 highlighting the divergent outcomes.

**Enhancing linear separability.** The first and rarer strategy, employed only by Pixtral, is to enhance linear separability using non-linear processes. This effectively continues the bottom-up perceptual work of the vision encoder. Its reasoning pathway processes the initial embeddings into a more linearly separable format. It refines features in a manner that aligns with the principles of contrastive learning (Alshammari et al., 2025). This results in a strong statistical alignment between its generative output and a linear probe on the improved final representations.

**Non-linear decision logic.** The vast majority of successes, however, were achieved by implementing a non-linear decision logic, a form of top-down reasoning that computes a solution not linearly readable from the visual embeddings themselves. It leverages the computational depth of the

Table 1: Performance metrics for selected SoTA models on the OpenWorld dataset, comparing direct and CoT generative accuracy with initial (LSC) and final representation linear separability accuracies.

| Model | Direct acc (%) | CoT acc (%) | LSC (%) | Linear separability (final, %) |
|---|---|---|---|---|
| Pixtral 12B | 79.4 | 84.2 | 76.0 | **88.0** |
| Gemma3 27B | 93.2 | 87.2 | **88.6** | collapsed (50.0) |
| Qwen2.5-VL 72B | 93.6 | 93.2 | **86.8** | 74.8 |
| Human | 91.0 | - | - | - |

model to form a non-linear decision boundary. Gemma3 represents an extreme case, where its final embeddings undergo a representational collapse with respect to linear separability (accuracy is 50%). This pathway highlights that a high LSC is a valuable starting point, but success also depends on a powerful non-linear process operating within a non-linear representational geometry.

**Statistical dependence of these pathways.** To better understand the connection between these reasoning pathways, we perform a $\chi^2$ test for statistical dependence between the generative predictions and the outcomes of our linear probes (Table 2). The generative output is positively correlated with linear separability probe on most models' vision and final embeddings, suggesting that more linearly separable representations generally lead to better generative performance. Interestingly, generative performance is **inversely** correlated with the linear separability of the *final* embeddings, implying the model is making a guess different from what it perceives to be the right guess. To move beyond this correlational ambiguity and identify the true drivers of performance, we first investigate the optimal intervention points for enhancing the model's generative reasoning, and then further explore this behavior.

Table 2: Summary of $\chi^2$ statistical dependence tests between generative performance and linear probe outcomes across 128 VLM configurations, where **(+)** and **(-)** denote significant positive/inverse dependence (both methods tend to agree/disagree), and **(Ø)** for no significant dependence.

| Probe location | + | - | Ø |
|---|---|---|---|
| Vision embeddings | 95 | 2 | 31 |
| Final embeddings | 74 | **29** | 25 |

## 5.2 Enhancing generative accuracy beyond LSC

To surpass the LSC, we compare three intervention types: at the vision-language projector, input-level guidance, and full model adaptation. Input guidance includes prompt tuning (trainable soft prompts with a meta-learning objective). Full adaptation uses LoRA, injecting trainable matrices into attention and MLP layers. We also test robustness to unseen prompt formats, with full details in Appendix G.

**Pinpointing the location for intervention.** An ablation study revealed that the intervention's location is key for surpassing the alignment gap. The results, summarized in Table 3, show that minimalist interventions at the vision-language interface lead to prompt overfitting rather than robust reasoning. We therefore froze the projector in all subsequent experiments. Our results pinpoint the bottleneck is in the LLM's reasoning pathways, not the vision encoder's perception. An ablation study confirmed this (Appendix J), as applying LoRA to the vision encoder resulted in nearly identical predictions and no additional performance gain compared to adapting the LLM alone. This suggests any adaptations within the vision encoder are likely a form of **naïve loss minimization** (Prieto et al., 2025), merely scaling existing features rather than learning more discriminative ones. The core issue is therefore not a perceptual deficit but a deeper failure in how the LLM processes its visual inputs.

Table 3: Phi model accuracies (%) on the OpenWorld dataset. The in-distribution (ID) evaluation uses the interleaved prompt structure seen during training. For the out-of-distribution (OOD) test, we use a labeled prompt structure that groups all images by category at the end of the prompt. In the generative columns, **bold** indicates the LSC is surpassed.

| Tuning method | ID | OOD | LSC |
|---|---|---|---|
| Direct baseline | 59.0 | 79.4 | 84.0 |
| Projector ($\mathcal{L}_{NT}$) | **90.2** | 73.8 | 84.0 |
| Postfix tuning ($\mathcal{L}_{NT}$) | **94.2** | **90.2** | 84.2 |
| Prompt tuning ($\mathcal{L}_{NT}$) | **90.4** | 83.4 | 84.2 |
| LoRA ($\mathcal{L}_{NT}$) | **92.2** | **89.8** | 84.4 |
| LoRA ($\mathcal{L}_{NT}$ llm-only) | **92.4** | **89.6** | 84.2 |

**Activation vs. adaptation.** With the bottleneck identified in the LLM's reasoning module, we find that targeted interventions can robustly unlock latent abilities. Our findings reveal a critical distinction between two approaches: **activating** latent skills versus **adapting** core weights, with the required method depending on the reasoning task. The success of postfix tuning, a methodological control that cannot alter visual representations, provides evidence that VLMs possess powerful, dormant reasoning pathways capable of performing **non-linear decision logic**, activating reasoning pathways.

As shown in Table 4, performance on OpenWorld is an **activation** issue. Prompt-based methods are effective, achieving performance comparable or even superior to LoRA. This suggests the model's inherent skills only need to be steered by an optimal input, a process similar to prompt engineering (Burns et al., 2023; Brown et al., 2020). On the relational HOI task, activation methods prove insufficient, requiring deeper **adaptation** via LoRA to substantially boost performance. This reveals a core principle, holding true for all tested models (Appendix K), that activation suffices for atomic semantic concept comparison while adaptation is required for reasoning over relational semantic concepts.

Table 4: Reported accuracies (%) for Phi model on both datasets, where **bold** indicates the LSC is surpassed.

| Method | OpenWorld | HOI |
|---|---|---|
| Direct baseline | 59.0 | 52.1 |
| LSC | 84.0 | 71.9 |
| Postfix tuning | **94.2** | 63.2 |
| LoRA ($\mathcal{L}_{\mathrm{NT}}$) | **92.2** | **78.6** |

## 6 CONNECTING THE TWO REASONING PATHWAYS

Our findings so far show that interventions using the standard $\mathcal{L}_{\mathrm{NT}}$ objective improve the **non-linear decision logic** pathway to surpass the LSC. This raises a question, however, what happens if we **combine** both reasoning pathways? Instead of forcing a non-linear decision logic to overcome poor separability, can we redirect these non-linear computational pathways to improve the linear separability of final embeddings also?

To test this, we introduce a combined training objective designed to do exactly that, consisting of two objectives: next-token prediction ($\mathcal{L}_{\mathrm{NT}}$), the standard language modeling objective of maximizing the likelihood of the ground-truth output tokens; and an explicit contrastive loss ($\mathcal{L}_{\mathrm{sim}}$) designed to identify a positive sample from a set of negative samples (van den Oord et al., 2019) by means of a weighted sum.

$$\mathcal{L}_{\mathrm{combined}} = w_n \mathcal{L}_{\mathrm{NT}} + w_c \mathcal{L}_{\mathrm{sim}}$$

This contrastive term explicitly encourages the final embeddings of a query image to be closer to its true category centroid than the alternative, as seen on Figure 4, detailed further in Appendix H.

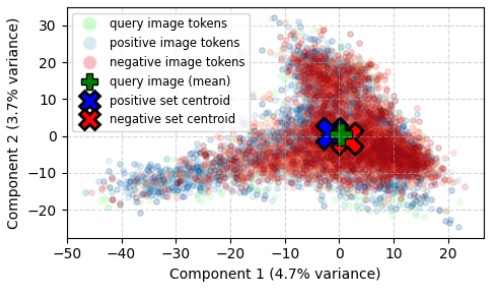 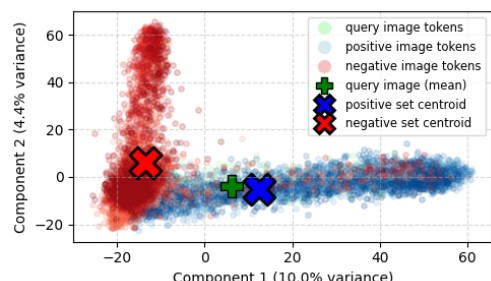

(a) Embedding space before.          (b) Embedding space after.

Figure 4: Principal component analysis (PCA) visualization of the embedding space before (a) and after (b) applying the contrastive term. As PCA applies only linear transformations, it preserves the relative geometric structure of the original space. In the plots, the green cross ($+$) is the query image's mean embedding, while the larger blue and red ($\mathbf{X}$) markers are the centroids for the positive and negative image sets, respectively. Before training, representations for positive (blue dots), negative (red dots), and query (green dots) images are heavily overlapped. After training with the contrastive loss, the space is restructured: the query image's representations (green dots) now occupy the same geometric region as the positive set (blue dots), clearly separating from the negative set (red dots).

We find that the balance between these objectives is critical; as shown in our sensitivity analysis in Appendix N, a dynamic dual cosine schedule results in a more controlled dynamic between the two loss counterparts.

## 6.1 COMPARING OBJECTIVES

Table 5 consolidates the performance of different PEFT strategies, comparing their generative and embedding similarity-based classification accuracy against relevant baselines.

Table 5: Summary of PEFT performance on Bongard tasks. In the generative columns, **bold** indicates the LSC is surpassed. Superscripts on separability scores denote a significant dependence ($p < 0.05$) with the predictions of the corresponding generative method (G).

| Model | Dataset | Method | Generative (%) | LSC (%) | Repr. acc. final (%) |
|---|---|---|---|---|---|
| Phi | OpenWorld | Direct baseline | 59.0 | $84.0^G$ | $76.4^{-G}$ |
| | | LoRA ($\mathcal{L}_{\text{NT}}$) | **92.2** | $84.4^G$ | 74.8 |
| | | LoRA ($\mathcal{L}_{\text{combined}}$) | **95.6** | $84.4^G$ | $93.8^G$ |
| | HOI | Direct baseline | 52.1 | 71.9 | $60.5^{-G}$ |
| | | LoRA ($\mathcal{L}_{\text{NT}}$) | **78.6** | $71.9^G$ | $63.6^G$ |
| | | LoRA ($\mathcal{L}_{\text{combined}}$) | **79.2** | $71.9^G$ | $82.0^G$ |
| Pixtral | OpenWorld | Direct baseline | 72.4 | $76.0^G$ | $87.2^G$ |
| | | LoRA ($\mathcal{L}_{\text{NT}}$) | **93.4** | 76.6 | $87.2^G$ |
| | | LoRA ($\mathcal{L}_{\text{combined}}$) | **95.0** | 74.4 | $96.2^G$ |
| | HOI | Direct baseline | 57.8 | $62.7^G$ | $70.2^G$ |
| | | LoRA ($\mathcal{L}_{\text{NT}}$) | **78.0** | $61.6^G$ | $74.9^G$ |
| | | LoRA ($\mathcal{L}_{\text{combined}}$) | **79.6** | $63.1^G$ | $77.8^G$ |
| Gemma3 4B | OpenWorld | Direct baseline | 76.0 | 89.8 | collapsed $(50.0)^G$ |
| | | LoRA ($\mathcal{L}_{\text{NT}}$) | 92.4 | $89.8^G$ | collapsed (50.0) |
| | | LoRA ($\mathcal{L}_{\text{combined}}$) | **95.6** | 89.8 | $96.6^G$ |
| | HOI | Direct baseline | 56.5 | 74.1 | collapsed $(50.0)^G$ |
| | | LoRA ($\mathcal{L}_{\text{NT}}$) | **84.2** | $74.1^G$ | collapsed (50.0) |
| | | LoRA ($\mathcal{L}_{\text{combined}}$) | **84.2** | $74.1^G$ | $83.2^G$ |
| Human | OpenWorld | | 91.0 | - | - |
| | HOI | | 91.4 | - | - |

**Aligned representational geometry.** Training with $\mathcal{L}_{\text{combined}}$ induces a significant and consistent statistical dependence between the generative and the final-layer linear probe predictions. Furthermore, their respective accuracies also converge. This convergence indicates a shift in the model's computational strategy. Its non-linear pathways are re-directed: instead of executing a non-linear decision logic on top of the representations, they are repurposed to actively refine the representations into a more linearly separable format based on surrounding context. We observe evidence for this in the model's internal attention mechanisms, where the most significant differences are concentrated in the later layers. This finding is consistent with research suggesting that reasoning mainly happens in middle-to-late layers, whereas perception is encoded in early layers (Chen et al., 2025). A visual analysis of these attention changes is available in Appendix P.

**Conceptual Generalization.** The performance gains from fine-tuning stem from genuine conceptual understanding rather than memorization. This is demonstrated on the Bongard-HOI dataset, where our models show robust generalization by maintaining high accuracy on test splits with entirely unseen objects and actions (Appendix L). This finding is reinforced by strong results on the OpenWorld test set, which consists entirely of new concepts by design. Whether further HOI progress needs better visual features or smarter reasoning pathways remains unclear.

**Generalization under domain shift.** Cross-domain evaluations reveal that the nature of the skill learned during fine-tuning is a key determinant of its transferability. For instance, the relational reasoning skills acquired from the HOI dataset transferred broadly across both our cross-Bongard task and the Winoground benchmark, whereas the atomic concept comparison skills from the OpenWorld dataset proved less generalizable (Appendix M). This distinction is particularly evident on Winoground, where only the HOI-trained models achieved substantial performance gains (Table 6). Here, our $\mathcal{L}_{\text{combined}}$ objective achieved the most successful transfer by consistently improving text retrieval accuracy for all models, a success we attribute to the inherent need for inter-image comparison in Bongard-style problems. Furthermore, this out-of-domain evaluation validated our LSC framework by demonstrating both its broader applicability and the difficulty of surpassing the LSC, which a fine-tuned Phi model could not significantly surpass.

| Model | Method | Acc. (%) |
|---|---|---|
| CLIP (ViT-L/14) | Baseline | **27.75** |
| Phi | Baseline | 16.75 |
| | $\mathcal{L}_{\text{NT}}$ | 18.25 |
| | $\mathcal{L}_{\text{combined}}$ | **29.25** |
| Pixtral | Baseline | 28.75 |
| | $\mathcal{L}_{\text{NT}}$ | 43.50 |
| | $\mathcal{L}_{\text{combined}}$ | **54.75** |
| Gemma3 4B | Baseline | 5.25 |
| | $\mathcal{L}_{\text{NT}}$ | 5.75 |
| | $\mathcal{L}_{\text{combined}}$ | **12.25** |

Table 6: Text retrieval on Winoground, comparing baselines with HOI-trained LoRAs. The CLIP model is included as an LSC baseline for the Phi model.

## 7 DISCUSSION, LIMITATIONS AND FUTURE-WORK

**Discussion.** Recent theoretical work establishes that transformer language models are almost surely **injective**, meaning they are structurally lossless and different inputs provably map to different internal representations (Nikolaou et al., 2025). The complete input information is therefore preserved. However, this injectivity is highly non-linear; the input undergoes a deep composition of attention, normalization, and activation functions. This creates a gap: while all information is likely *present*, it is not necessarily *accessible*. From a computational perspective, a linear representation is more likely to generalize to downstream tasks as it is more amenable to information extraction.

However, models are typically trained with a next-token prediction loss only, which does not explicitly enforce linear separability. Consequently, when models are subjected to instruction-tuning or reinforcement learning, the task-specific learning signal can cause representational degradation, where linearly decodable information is diminished. The LSC framework detects this degradation, while the auxiliary contrastive signal actively restores and promotes linear separability. As visualized in Figure 5, this signal restructures the manifold into *one-dimensionally linear* "rays" (further detailed in Appendix B), effectively structuring representations into globally consistent semantic directions.

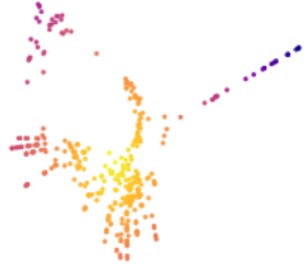

Figure 5: Manifold structure.

**Limitations.** While $\mathcal{L}_{\text{combined}}$ objective successfully restructures vision representations, it appears to compromise the complex, non-linear decision logic built upon them, effectively reintroducing an image-to-text alignment gap. This brittleness became apparent when testing model robustness on the HOI dataset (Appendix K). Notably, all three models trained with $\mathcal{L}_{\text{combined}}$ suffered performance degradation when the prompt format was altered, whereas only one model trained with $\mathcal{L}_{\text{NT}}$ objective exhibited a similar decline. This pattern extends to multiple-choice VQA benchmarks, where models trained with the $\mathcal{L}_{\text{combined}}$ objective experienced steeper performance drops compared to their $\mathcal{L}_{\text{NT}}$ counterparts (Appendix A).

**Future work.** It is inherent that significantly altering the geometry of representations disrupts the reasoning pathways calibrated to them. To mitigate this, we propose a promising avenue for future work: enforcing **image-to-text alignment** where the one-dimensional linearity observed in our visual manifold is also explicitly promoted within text representations. By anchoring semantically analogous visual embeddings to this structured textual space, we can resolve the "split-brain" pathology of current VLMs — characterized by disjoint heuristics for visual and textual processing — thereby establishing a foundation for unified, truly multimodal reasoning.

## 8 CONCLUSION

This work identifies a pervasive "alignment gap" in VLMs, where reasoning capabilities often lag behind the linear separability of their own visual perceptions on Bongard tasks. By introducing the Linear Separability Ceiling (LSC), we demonstrate that this bottleneck on baseline models is not stemming from poor perception but from weak top-down reasoning.

Our key contributions include:
- a novel diagnostic framework utilizing **prototype vectors** (centroids) to establish the LSC, which provides a geometric and statistical lens to formalize and quantify the alignment gap;
- the identification of two distinct pathways to surpass the LSC: enhancing linear separability or executing non-linear decision logic — where we show that the latter can be effectively activated via **postfix tuning**;
- a training methodology employing an **auxiliary contrastive objective** with a **dual cosine schedule**, which successfully balances the conflicting signals of representational structuring and next-token prediction;
- and empirical evidence linking the geometry of **one-dimensionally linear representations** to improved benchmark performance, supporting the hypothesis that globally consistent structures encourage transfer to out-of-distribution tasks.

Our work provides both validated methods for enhancing VLM reasoning and novel insights into the representational properties of multimodal models, paving the way for more capable and interpretable AI.

## 9 LLM USAGE

The authors utilized a LLM to refine grammar and sentence structure, and to aid in discovering related work. The authors reviewed all generated content and assume full responsibility for this publication.

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

## A  LSC ON VQA

To show that LSC is useful to disentangle perception from reasoning beyond Bongard tasks, we evaluate the Phi-3.5 model, its respective vision encoder (CLIP ViT-L/14) and its respective text encoder on multiple choice VQA tasks:

- **Bongard-HOI** and **POPE** (Li et al., 2023): Converted from binary to multiple-choice tasks operating on a single image. Each sample was restructured to contain one correct answer and three distractors (4 options total).
- **A-OKVQA** (Schwenk et al., 2022): A VQA task requiring world knowledge and common-sense reasoning.
- **ScienceQA** (Lu et al., 2022): A multimodal benchmark evaluating reasoning capabilities.

We established the LSC for these VQA tasks using the CLIP vision and text encoder. We compared this against a "random" baseline (simulating a model that always selects random answer), the standard Phi-3.5 baseline (zero-shot) and $\mathcal{L}_{NT}$ and $\mathcal{L}_{combined}$ fine-tuned models on Bongard HOI dataset. The results are summarized in Table A.1.

Table A.1: LSC analysis across VQA benchmarks. LSC (%) is the accuracy of a linear probe on the CLIP vision and text encoder. Random (%) denotes the accuracy when always selecting answer randomly. Gen. (%) is the performance of the Phi-3.5 model when prompted zero-shot without CoT. We compare the baseline Phi model to $\mathcal{L}_{NT}$ and $\mathcal{L}_{combined}$ fine-tuned models on Bongard HOI dataset.

| Dataset | LSC (%) | Random (%) | Baseline gen. (%) | $\mathcal{L}_{NT}$ gen. (%) | $\mathcal{L}_{combined}$ gen. (%) |
| --- | --- | --- | --- | --- | --- |
| HOI | 78.75 | 25.04 | 78.89 | 67.57 | 54.53 |
| POPE | 62.53 | 25.00 | 79.65 | 71.34 | 68.40 |
| A-OKVQA | 61.14 | 24.54 | 74.34 | 64.72 | 55.73 |
| ScienceQA | 46.21 | 35.35 | 87.42 | 79.68 | 73.84 |

**Conclusion.** The scores between the LSC and the baseline model being identical for HOI dataset indicates the model is extracting all the linearly available information. For POPE, however, even when operating on the exact same COCO dataset images, the baseline model achieved results significantly higher than the LSC. This gap suggests two possibilities: it may be an artifact of image-concept memorization by Phi during its training, or it indicates that the necessary representations are inherently non-linear, requiring non-linear decision logic to be untangled. A-OKVQA and ScienceQA show the largest divergence, where the language-based abstract reasoning required exceeds the linear separability of the visual features alone. Furthermore, the effect of representational restructuring is most apparent on HOI and A-OKVQA, where the performance dropped significantly more with $\mathcal{L}_{combined}$ when compared to $\mathcal{L}_{NT}$. In contrast, POPE and ScienceQA are more resilient to this geometric shift. This suggests that the non-linear decision logic for these tasks relies less on representational geometry and more on memorized image-concept pairs, which remain robust to the imposed structural changes.

## B  VISUALIZING THE MANIFOLD STRUCTURE

Isomap is a non-linear dimensionality reduction technique that estimates the intrinsic geometry of a data manifold by preserving geodesic distances. While our PCA analysis demonstrated global linear separability, Figure A.1 reveals that the $\mathcal{L}_{combined}$ objective specifically promotes *one-dimensionally linear* representations, visualized here as distinct "rays" that persist across increasing neighborhood sizes ($k$). Furthermore, this geometric structure suggests that while semantic concepts are encoded as linear features (directions), the process of determining *which* linear axis is relevant for a specific query is context-dependent, necessitating the execution of non-linear decision logic to select the correct linear probe.

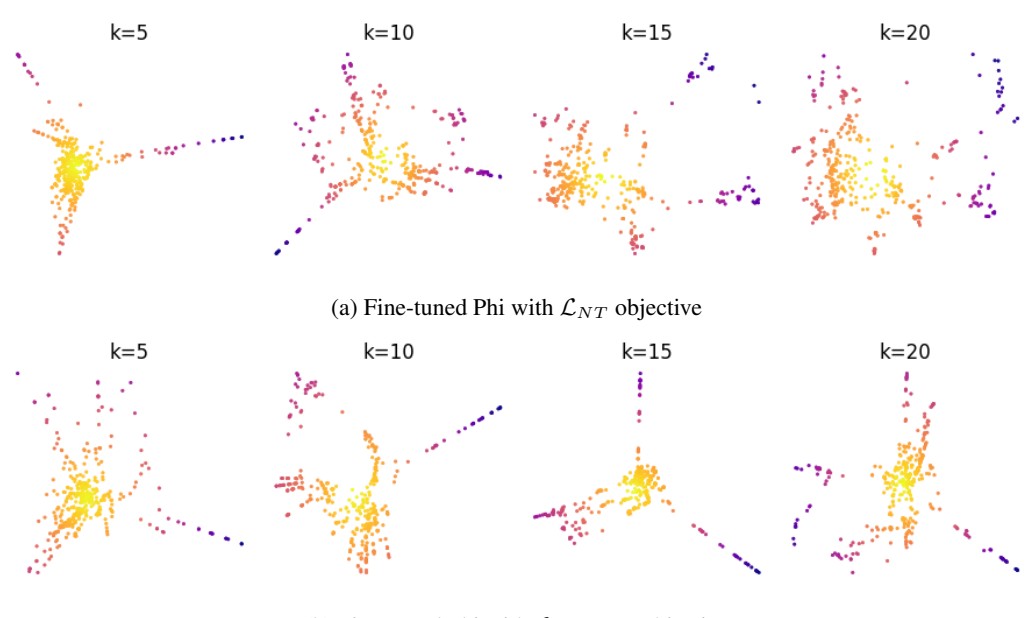

(a) Fine-tuned Phi with $\mathcal{L}_{NT}$ objective

(b) Fine-tuned Phi with $\mathcal{L}_{combined}$ objective

Figure A.1: Isomap projections of the positive concept centroids from the test set, colored by distance from the geometric center. **(a)** The $\mathcal{L}_{NT}$ trained model shows a manifold that loses structure as the neighborhood size ($k$) increases. **(b)** The model trained with $\mathcal{L}_{combined}$ reveals a manifold that is more robust as the neighborhood size ($k$) increases. The persistence of these "rays" at higher $k$ values indicates that semantic concepts are globally more consistent.

# C   BASELINE PROMPT DEFINITIONS

Strategies evaluated in Table A.3 varied primarily in how images were presented relative to descriptive text, the position of the query image, and whether CoT reasoning was asked.

**Interleaved strategy.**

```
You are presented a Bongard task. There are {image_count} pictures total.
First {cat_imgs} samples belong to cat_2, which follow 1 common rule. Here they are: {cat2_imgs}.
Following {cat_imgs} distinctly do not follow that rule and are cat_1. Here they are: {cat1_imgs}.
Last image is a query image you need to categorize either as cat_2 or cat_1 based on the rule,
which is here: {query_img}.
If it follows the rule, it's cat_2. If it doesn't follow the rule, it's cat_1.
```

**Interleaved query first strategy.**

```
You are presented a Bongard task. There are {image_count} pictures total.
First image is a query image you need to categorize either as cat_2 or cat_1 based on the rule.
Here is a query image: {query_img}.
{cat_imgs} samples belong to cat_2, which follow 1 common rule. Here they are: {cat2_imgs} .
Following {cat_imgs} distinctly do not follow that rule and are cat_1. Here they are: {cat1_imgs} .
If query image follows the rule, it's cat_2. If it doesn't follow the rule, it's cat_1.
```

**Labeled strategy.**

```
You are presented a Bongard task. There are {image_count} pictures total.
First {cat_imgs} samples belong to cat_2, which follow 1 common rule.
Following {cat_imgs} distinctly do not follow that rule and are cat_1.
Last image is a query image you need to categorize either as cat_2 or cat_1 based on the rule.
If query image follows the rule, it's cat_2. If it doesn't follow the rule, it's cat_1.
Here are the cat2 images: {cat2_imgs} , cat1 images: {cat1_imgs} , query image: {query_img} .
```

**Labeled query first strategy.**

```
You are presented a Bongard task. There are {image_count} pictures total.
First image is a query image you need to categorize either as cat_2 or cat_1 based on the rule.
Following {cat_imgs} samples belong to cat_2, which follow 1 common rule.
Last {cat_imgs} distinctly do not follow that rule and are cat_1.
If query image follows the rule, it's cat_2. If it doesn't follow the rule, it's cat_1.
Here is the query image: {query_img} , cat2 images: {cat2_imgs} , cat1 images: {cat1_imgs}.
```

One of the following string variables is appended to the output of the core prompt.

**Direct conclusion prompt string.**

```
Your task is to:
1. Provide your conclusion for the 'query image' if it can be categorized as
either 'cat_1' or 'cat_2' based on the analysis and the rule.

The format of your output should be as follows:
Conclusion: cat_1 or cat_2

Conclusion should be 1 category only without extra symbols!
```

**CoT prompt string.**

```
Your task is to:
1. Determine the rule or criterion that distinguishes the 'cat_2' samples from the 'cat_1' ones.
2. Analyse the 'query image' (last image).
3. Provide your conclusion for the 'query image' if it can be categorized as
either 'cat_1' or 'cat_2' based on the analysis and the rule.

Ensure that the output is clear, well-formatted, and free of unnecessary explanations.
The format of your output should be as follows:
Analysis: (Your analysis here)
Rule: (The distinguishing rule here)
query image: (query image details)
Conclusion: cat_1 or cat_2

Conclusion should be 1 category only without extra symbols!
```

## D    METRIC EXTRACTION METHODOLOGY

As illustrated in Figure A.2, performance metrics are derived from three distinct architectural stages. LSC is calculated by applying a nearest-centroid linear probe to the raw, non-contextualized embeddings directly from the vision encoder. The final representation accuracy (acc final) utilizes the same probe on the contextualized hidden states extracted from the last layer of the LLM. The generative accuracy (acc gen) is determined by evaluating the model's standard next-token textual prediction against the ground truth.

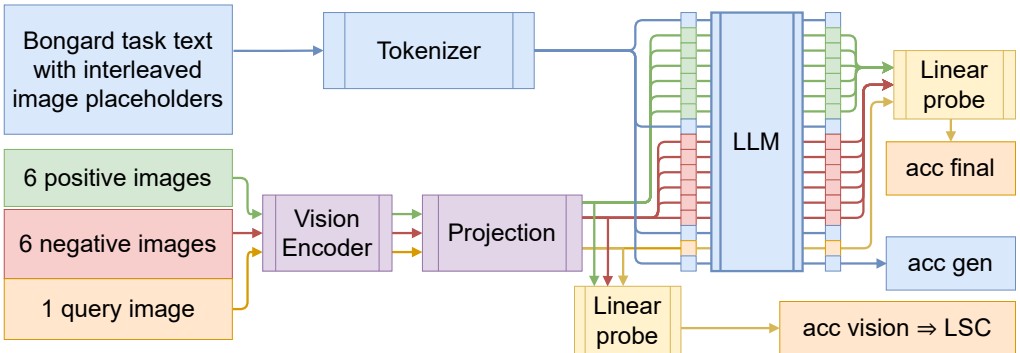

Figure A.2: Flowchart illustrating the extraction points for visual embeddings and generative outputs within the VLM architecture.

# E SINGLE VS BATCHED CONTEXT

We performed a preliminary analysis comparing a nearest-centroid classifier approach against a simpler nearest-neighbor classifier. This allows us to demonstrate that aggregating visual information into a conceptual prototype is more effective than treating each piece of visual evidence in isolation. We refer to these two distinct classification strategies as: **Single context**, a nearest-neighbor approach where the query image is classified based on its cosine similarity to the single closest example image vector from the combined positive and negative sets. This method is sensitive to the features of individual, and potentially outlier, examples; **Batched context**, a nearest-centroid approach selected for the LSC. The query image is classified based on its cosine similarity to a prototype vector (centroid) representing the entire set of positive or negative examples. This method is designed to capture the underlying abstract rule of the set. The results of this comparison are presented in Table A.2.

Table A.2: Comparison of classification accuracy for single vs. batched context.

| Model | Dataset | Context | Sim. acc (vision, %) | Sim. acc (final, %) |
|---|---|---|---|---|
| Phi | OpenWorld | Single | $78.8 \pm 3.6$ | $78.8 \pm 3.6$ |
| | | Batched | $84.0 \pm 3.2$ | $87.0 \pm 2.9$ |
| | HOI | Single | $64.9 \pm 3.3$ | $66.4 \pm 3.3$ |
| | | Batched | $71.9 \pm 3.1$ | $69.6 \pm 3.2$ |
| Pixtral | OpenWorld | Single | $70.4 \pm 4.0$ | $78.6 \pm 3.6$ |
| | | Batched | $76.0 \pm 3.7$ | $88.2 \pm 2.8$ |
| | HOI | Single | $62.4 \pm 3.4$ | $68.0 \pm 3.2$ |
| | | Batched | $62.9 \pm 3.3$ | $72.4 \pm 3.1$ |
| Gemma3 4B | OpenWorld | Single | $77.8 \pm 3.6$ | $81.6 \pm 3.4$ |
| | | Batched | $89.6 \pm 2.7$ | $76.6 \pm 3.7$ |
| | HOI | Single | $69.1 \pm 3.2$ | $69.8 \pm 3.2$ |
| | | Batched | $74.1 \pm 3.0$ | $70.2 \pm 3.2$ |
| Gemma3 27B | OpenWorld | Single | $78.2 \pm 3.6$ | $78.4 \pm 3.6$ |
| | | Batched | $88.6 \pm 2.8$ | $78.8 \pm 3.6$ |
| | HOI | Single | $68.4 \pm 3.2$ | $69.8 \pm 3.2$ |
| | | Batched | $73.4 \pm 3.1$ | $64.8 \pm 3.3$ |
| InternVL 14B | OpenWorld | Single | $69.4 \pm 4.0$ | $74.0 \pm 3.8$ |
| | | Batched | $70.2 \pm 4.0$ | $78.4 \pm 3.6$ |
| | HOI | Single | $63.7 \pm 3.3$ | $64.4 \pm 3.3$ |
| | | Batched | $65.1 \pm 3.3$ | $68.0 \pm 3.2$ |
| Qwen2.5-VL 7B | OpenWorld | Single | $81.0 \pm 3.4$ | $75.4 \pm 3.8$ |
| | | Batched | $88.2 \pm 2.8$ | $81.2 \pm 3.4$ |
| | HOI | Single | $69.1 \pm 3.2$ | $66.8 \pm 3.3$ |
| | | Batched | $72.1 \pm 3.1$ | $65.9 \pm 3.3$ |
| Qwen2.5-VL 72B | OpenWorld | Single | $79.0 \pm 3.6$ | $71.2 \pm 4.0$ |
| | | Batched | $87.0 \pm 2.9$ | $70.2 \pm 4.0$ |
| | HOI | Single | $69.2 \pm 3.2$ | $60.6 \pm 3.4$ |
| | | Batched | $72.0 \pm 3.1$ | $58.5 \pm 3.4$ |

The data indicates that the batched context method provides a significantly more accurate and stable measure of linear separability. By averaging embeddings to create a conceptual prototype, the classifier becomes more resilient to individual outlier examples and better reflects the abstract visual rule. For this reason, all LSC scores reported in the main body of this paper are calculated using this batched context methodology.

# F  SEPARABILITY CEILING IN VLMS

Table A.3: Generative performance vs. the LSC and the same linear probe accuracy on its final representations. Superscripts on embedding scores denote significant (p<0.05) dependence (negative indicating inverse) with generative predictions (D=Direct, C=CoT).

| Model | Dataset | Prompt Strategy | Direct acc (%) | CoT acc (%) | LSC | linear probe (final, %) |
|---|---|---|---|---|---|---|
| Phi | OpenWorld | Interleaved | $59.0 \pm 4.3$ | $80.6 \pm 3.5$ | $84.0 \pm 3.2^{D}$ | $76.4 \pm 3.7^{-D}$ |
| | | Interleaved query first | $52.4 \pm 4.4$ | $68.0 \pm 4.1$ | $84.0 \pm 3.2^{D}$ | $66.0 \pm 4.2^{D,C}$ |
| | | Labeled | $79.4 \pm 3.5$ | $78.8 \pm 3.6$ | $84.0 \pm 3.2$ | $78.2 \pm 3.6$ |
| | | Labeled query first | $57.2 \pm 4.3$ | $64.0 \pm 4.2$ | $84.0 \pm 3.2^{D}$ | $65.0 \pm 4.2^{D,C}$ |
| | HOI | Interleaved | $52.1 \pm 3.5$ | $59.9 \pm 3.4$ | $71.9 \pm 3.1^{C}$ | $60.5 \pm 3.4^{-D}$ |
| | | Interleaved query first | $51.5 \pm 3.5$ | $55.5 \pm 3.4$ | $71.9 \pm 3.1^{D,C}$ | $56.5 \pm 3.4^{D,C}$ |
| | | Labeled | $58.8 \pm 3.4$ | $57.4 \pm 3.4$ | $71.9 \pm 3.1^{C}$ | $60.6 \pm 3.4^{-C}$ |
| | | Labeled query first | $55.1 \pm 3.4$ | $55.8 \pm 3.4$ | $71.9 \pm 3.1^{D}$ | $56.1 \pm 3.4^{D,C}$ |
| Pixtral | OpenWorld | Interleaved | $72.4 \pm 3.9$ | $83.8 \pm 3.2$ | $76.0 \pm 3.7^{D,C}$ | $87.2 \pm 2.9^{D,C}$ |
| | | Interleaved query first | $71.2 \pm 4.0$ | $81.0 \pm 3.4$ | $75.8 \pm 3.8^{D,C}$ | $76.2 \pm 3.7^{C}$ |
| | | Labeled | $79.4 \pm 3.5$ | $84.2 \pm 3.2$ | $76.0 \pm 3.7^{D}$ | $88.0 \pm 2.8^{D,C}$ |
| | | Labeled query first | $62.4 \pm 4.2$ | $76.0 \pm 3.7$ | $75.8 \pm 3.8^{D,C}$ | $75.0 \pm 3.8^{D,C}$ |
| | HOI | Interleaved | $57.8 \pm 3.4$ | $66.9 \pm 3.3$ | $62.7 \pm 3.4^{D,C}$ | $70.2 \pm 3.2^{D,C}$ |
| | | Interleaved query first | $53.5 \pm 3.5$ | $67.2 \pm 3.3$ | $63.2 \pm 3.3^{C}$ | $62.6 \pm 3.4^{-D,C}$ |
| | | Labeled | $60.6 \pm 3.4$ | $66.6 \pm 3.3$ | $62.7 \pm 3.4^{D,C}$ | $70.5 \pm 3.2^{C}$ |
| | | Labeled query first | $53.4 \pm 3.5$ | $65.4 \pm 3.3$ | $63.2 \pm 3.3^{C}$ | $61.1 \pm 3.4^{D,C}$ |
| Gemma3 4B | OpenWorld | Interleaved | $76.0 \pm 3.7$ | $80.2 \pm 3.5$ | $89.8 \pm 2.7^{C}$ | $50.0^{D,-C}$ |
| | | Interleaved query first | $50.2 \pm 4.4$ | $51.4 \pm 4.4$ | $89.8 \pm 2.7^{D,C}$ | $50.0^{D,C}$ |
| | | Labeled | $68.6 \pm 4.1$ | $83.0 \pm 3.3$ | $89.8 \pm 2.7^{C}$ | $50.0^{D}$ |
| | | Labeled query first | $52.0 \pm 4.4$ | $52.4 \pm 4.4$ | $89.8 \pm 2.7^{D,C}$ | $50.0^{D,C}$ |
| | HOI | Interleaved | $56.5 \pm 3.4$ | $62.6 \pm 3.4$ | $74.1 \pm 3.0^{C}$ | $50.0^{D,-C}$ |
| | | Interleaved query first | $50.4 \pm 3.5$ | $50.7 \pm 3.5$ | $74.1 \pm 3.0^{D}$ | $50.0^{D,C}$ |
| | | Labeled | $54.2 \pm 3.5$ | $63.9 \pm 3.3$ | $74.1 \pm 3.0^{C}$ | $50.0^{D,-C}$ |
| | | Labeled query first | $50.7 \pm 3.5$ | $51.7 \pm 3.5$ | $74.1 \pm 3.0$ | $50.0^{D,C}$ |
| Gemma3 27B | OpenWorld | Interleaved | $91.8 \pm 2.4$ | $88.6 \pm 2.8$ | $88.6 \pm 2.8^{D,C}$ | $50.0^{-D,C}$ |
| | | Interleaved query first | $60.6 \pm 4.3$ | $71.4 \pm 4.0$ | $88.6 \pm 2.8^{D,C}$ | $50.0^{D,C}$ |
| | | Labeled | $93.2 \pm 2.2$ | $87.2 \pm 2.9$ | $88.6 \pm 2.8^{D}$ | $50.0^{C}$ |
| | | Labeled query first | $54.6 \pm 4.4$ | $67.2 \pm 4.1$ | $88.6 \pm 2.8^{D,C}$ | $50.0^{D,C}$ |
| | HOI | Interleaved | $75.1 \pm 3.0$ | $75.2 \pm 3.0$ | $73.4 \pm 3.1^{D,C}$ | $50.0^{-D,C}$ |
| | | Interleaved query first | $54.4 \pm 3.5$ | $65.0 \pm 3.3$ | $73.4 \pm 3.1^{D,C}$ | $50.0^{D,C}$ |
| | | Labeled | $76.2 \pm 2.9$ | $75.1 \pm 3.0$ | $73.4 \pm 3.1^{D,C}$ | $50.0^{C}$ |
| | | Labeled query first | $52.2 \pm 3.5$ | $59.6 \pm 3.4$ | $73.4 \pm 3.1^{D,C}$ | $50.0^{D,C}$ |
| InternVL 14B | OpenWorld | Interleaved | $79.4 \pm 3.5$ | $80.6 \pm 3.5$ | $70.2 \pm 4.0$ | $61.6 \pm 4.3$ |
| | | Interleaved query first | $80.4 \pm 3.5$ | $70.4 \pm 4.0$ | $70.2 \pm 4.0$ | $59.4 \pm 4.3^{C}$ |
| | | Labeled | $55.8 \pm 4.4$ | $71.8 \pm 3.9$ | $70.2 \pm 4.0^{D}$ | $62.4 \pm 4.2^{-D}$ |
| | | Labeled query first | $66.6 \pm 4.1$ | $73.0 \pm 3.9$ | $70.2 \pm 4.0^{D,C}$ | $54.0 \pm 4.4^{D,C}$ |
| | HOI | Interleaved | $58.2 \pm 3.4$ | $63.6 \pm 3.3$ | $65.1 \pm 3.3^{D,C}$ | $52.0 \pm 3.5^{-D}$ |
| | | Interleaved query first | $64.2 \pm 3.3$ | $60.4 \pm 3.4$ | $65.1 \pm 3.3^{D,C}$ | $52.8 \pm 3.5^{D,C}$ |
| | | Labeled | $52.0 \pm 3.5$ | $58.8 \pm 3.4$ | $65.1 \pm 3.3^{D}$ | $53.1 \pm 3.5^{-D}$ |
| | | Labeled query first | $56.5 \pm 3.4$ | $62.7 \pm 3.4$ | $65.1 \pm 3.3^{D,C}$ | $51.5 \pm 3.5^{D,C}$ |
| InternVL3 78B | OpenWorld | Interleaved | $73.0 \pm 3.9$ | $69.9 \pm 4.1$ | $75.6 \pm 3.8^{C}$ | $72.4 \pm 3.9^{-C}$ |
| | | Interleaved query first | $88.4 \pm 2.8$ | $79.4 \pm 3.8$ | $75.6 \pm 3.8^{D,C}$ | $69.8 \pm 4.0^{C}$ |
| | | Labeled | $64.4 \pm 4.2$ | $67.9 \pm 6.7$ | $75.6 \pm 3.8$ | $73.4 \pm 3.9^{-D}$ |
| | | Labeled query first | $84.6 \pm 3.2$ | $76.6 \pm 4.8$ | $75.6 \pm 3.8^{D,C}$ | $67.2 \pm 4.1^{D,C}$ |
| | HOI | Interleaved | $59.0 \pm 3.4$ | $62.1 \pm 3.4$ | $66.1 \pm 3.3^{D,C}$ | $57.6 \pm 3.4^{-C}$ |
| | | Interleaved query first | $66.1 \pm 3.3$ | $62.0 \pm 3.4$ | $66.1 \pm 3.3^{D,C}$ | $58.6 \pm 3.4^{-D,C}$ |
| | | Labeled | $54.0 \pm 3.5$ | $60.4 \pm 3.7$ | $66.1 \pm 3.3^{D}$ | $60.4 \pm 3.4^{-D,-C}$ |
| | | Labeled query first | $66.5 \pm 3.3$ | $63.0 \pm 3.5$ | $66.1 \pm 3.3^{D}$ | $57.1 \pm 3.4^{D,C}$ |
| Qwen2.5-VL 7B | OpenWorld | Interleaved | $86.4 \pm 3.0$ | $84.6 \pm 3.2$ | $87.8 \pm 2.9^{D,C}$ | $58.6 \pm 4.3^{-D,-C}$ |
| | | Interleaved query first | $51.2 \pm 4.4$ | $78.2 \pm 3.6$ | $87.8 \pm 2.9^{-D,C}$ | $54.0 \pm 4.4^{-D,C}$ |
| | | Labeled | $92.2 \pm 2.4$ | $87.2 \pm 2.9$ | $87.8 \pm 2.9^{D,C}$ | $57.0 \pm 4.3^{-C}$ |
| | | Labeled query first | $54.8 \pm 4.4$ | $80.6 \pm 3.5$ | $87.8 \pm 2.9^{-D,C}$ | $54.8 \pm 4.4^{-D,C}$ |
| | HOI | Interleaved | $63.0 \pm 3.3$ | $67.4 \pm 3.2$ | $72.2 \pm 3.1^{D,C}$ | $52.2 \pm 3.5^{-D,-C}$ |
| | | Interleaved query first | $50.4 \pm 3.5$ | $60.8 \pm 3.4$ | $72.2 \pm 3.1^{C}$ | $51.1 \pm 3.5^{-D,C}$ |
| | | Labeled | $70.4 \pm 3.2$ | $68.4 \pm 3.2$ | $72.2 \pm 3.1^{D,C}$ | $50.5 \pm 3.5^{D,-C}$ |
| | | Labeled query first | $50.6 \pm 3.5$ | $63.0 \pm 3.3$ | $72.2 \pm 3.1^{C}$ | $51.7 \pm 3.5^{-D,C}$ |
| Qwen2.5-VL 72B | OpenWorld | Interleaved | $93.6 \pm 2.1$ | $89.2 \pm 2.7$ | $86.8 \pm 3.0^{D,C}$ | $59.0 \pm 4.3$ |
| | | Interleaved query first | $91.6 \pm 2.4$ | $91.8 \pm 2.4$ | $86.8 \pm 3.0^{D,C}$ | $68.2 \pm 4.1^{D,C}$ |
| | | Labeled | $93.6 \pm 2.1$ | $93.2 \pm 2.2$ | $86.8 \pm 3.0^{D,C}$ | $74.8 \pm 3.8^{D,C}$ |
| | | Labeled query first | $91.8 \pm 2.4$ | $92.2 \pm 2.4$ | $86.8 \pm 3.0^{D,C}$ | $65.2 \pm 4.2^{C}$ |
| | HOI | Interleaved | $79.9 \pm 2.8$ | $78.5 \pm 2.8$ | $72.1 \pm 3.1^{D,C}$ | $53.9 \pm 3.5^{-C}$ |
| | | Interleaved query first | $77.1 \pm 2.9$ | $77.2 \pm 2.9$ | $72.1 \pm 3.1^{D,C}$ | $59.4 \pm 3.4^{D,C}$ |
| | | Labeled | $80.2 \pm 2.8$ | $79.2 \pm 2.8$ | $72.1 \pm 3.1^{D,C}$ | $63.7 \pm 3.3^{D,C}$ |
| | | Labeled query first | $76.0 \pm 3.0$ | $77.5 \pm 2.9$ | $72.1 \pm 3.1^{D,C}$ | $56.2 \pm 3.4^{D,C}$ |

## G  FINE-TUNING DETAILS

**LoRA and projector tuning** employed a learning rate of $1e^{-4}$ and a batch size of 25. Using LoRA we inject trainable matrices into the attention and MLP weights. LoRA experiments utilized a rank of $r = 8$ and a scaling factor of $\alpha = 8$. These hyperparameters are consistent with established practices for fine-tuning VLMs and training LoRA adapters (Hu et al., 2022; Liu et al., 2023; Microsoft, 2025).

**Prompt tuning** method reported in the main paper employs a hybrid approach, prepending and appending learnable soft prompt embeddings to the input sequence concurrently. The final structure consists of a prefix prompt, 100 learnable embeddings prepended to the standard interleaved prompt. And instructional postfix prompt, a fixed, human-readable instruction designed to encourage meta-learning, followed by 100 learnable embeddings.
The fixed instruction text is:

```
Also construct a specific instruction which you think will help the most for this kind of task.
Think of an algorithm to follow, what kind of components to look for on the images,
how they could be combined to follow some rule. Be creative! :D
Return the classification before anything else
and then the specific instruction word-by-word and nothing else!

Here is the specific instruction which helps you a lot:
```

**Postfix tuning** method employs postfix part of prompt tuning only.

**Hyperparameter search** revealed that an aggressive configuration with a small batch size (2) and a high learning rate (0.01), was highly effective for postfix tuning. Prompt tuning uses a learning rate of $1e^{-4}$ and a batch size of 25.

**Evaluation.** To evaluate structural robustness, we test performance on unseen prompt formats. The in-distribution (ID) evaluation uses the standard interleaved prompt structure seen during training. For the out-of-distribution (OOD) test, we use a labeled prompt structure that groups all images by category at the end of the prompt. This OOD setup challenges how well the optimization targets generalize beyond the syntactic structure on which they were trained. Furthermore, OpenWorld test set consists entirely of new concepts unseen in train set, verifying conceptual generation.

## H    COMBINED LOSS DETAILS

Inspired by the effectiveness of similarity measures in assessing embedding quality (Section 4), we explored an additional objective for tuning. Instead of relying solely on the standard next-token prediction loss ($\mathcal{L}_{\text{NT}}$), we incorporated a similarity-based contrastive loss ($\mathcal{L}_{\text{sim}}$) designed to directly enhance the discriminability of the final embeddings with respect to the task structure during tuning.

Specifically, let $V_T$ be the normalized mean-pooled embedding vector derived from the final hidden states corresponding to the query image $T$. Similarly, let $C_P$ and $C_N$ be the normalized mean-pooled centroids derived from the concatenated final hidden states of all images in the positive set $P = \{p_1, ..., p_k\}$ and negative set $N = \{n_1, ..., n_k\}$, respectively. These embeddings are obtained after the model processes the full input sequence, including the tunable soft prompt tokens. We calculate the cosine similarities between the query image embedding and the set centroids: $s_P = V_T \cdot C_P$ and $s_N = V_T \cdot C_N$.

The similarity-based loss, $\mathcal{L}_{\text{sim}}$, is then formulated as a cross-entropy loss over these similarities, akin to InfoNCE, encouraging the query image embedding to be closer to the centroid of its true category set:

$$\mathcal{L}_{\text{sim}} = \text{CrossEntropy}([\frac{s_P}{\tau}, \frac{s_N}{\tau}], y)$$

where $\tau$ (fixed to 0.07) is a temperature hyperparameter scaling the logits, and $y$ is the target label ($y = 0$ if the ground truth category for $T$ is the positive set $P$, and $y = 1$ if it is the negative set $N$). This contrastive loss term was combined with the standard next-token prediction loss, weighted by hyperparameters $w_n$ and $w_c$:

$$\mathcal{L}_{\text{combined}} = w_n \mathcal{L}_{\text{NT}} + w_c \mathcal{L}_{\text{sim}}$$

The gradients from this combined loss $\mathcal{L}_{\text{combined}}$ were used to update the parameters. This approach directly optimizes final embeddings to be well-clustered according to the Bongard task's positive/negative distinction, complementing the language modeling objective.

The loss counterparts can be conflicting and pull weights in different directions, and to achieve consistent results, for Bongard task training we exercised a separate schedule for both parts.

## I COMBINED LOSS HYPERPARAMETER SEARCH

To find hyperparameters for $\mathcal{L}_{\text{combined}} = w_n \mathcal{L}_{\text{NT}} + w_c \mathcal{L}_{\text{sim}}$, we fixed $w_n$ to 1 and scanned through $w_c$ values with different schedules. The constant schedule had the same $w_c$ throughout different epochs, which proved unstable for larger values. The linear schedule had a target value where every epoch the $w_c = \frac{1+\text{cur\_epoch}}{\text{total\_epochs}} \cdot \text{target\_value}$. Using linear schedule, scan was conducted over [0.1, 0.2, 0.4, 0.8, 1.6, 2.4, 3.2] target values. For Phi and Gemma3 the best value was 0.4 and for Pixtral best value was 1.6, which is shown on Figure A.3.

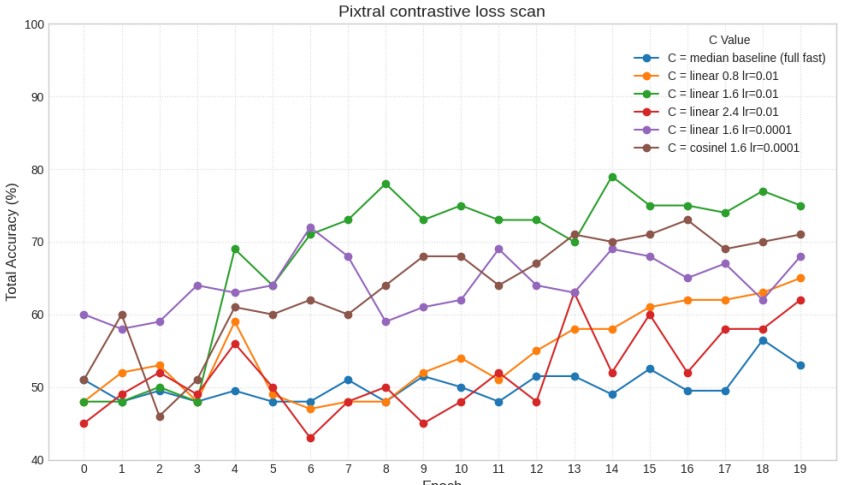

Figure A.3: Illustration of contrastive loss hyperparameter scan results for Pixtral. Plot displays model accuracy after every epoch (0-indexed) for selected hyperparameters.

Afterwards, target values were tested with different learning rates and since they proved stable, we picked these values. To ensure a more controlled dynamic where the contrastive loss provides an initial strong signal that gradually yields to the NTP objective for final model refinement, we introduced a cosine learning schedule which has two separate schedules for $w_n$ and $w_c$, governed by the overall training progress. This way we reduce conflicting learning signals and ensure similar results across different seeds. Let $p = \frac{\text{current\_step}+1}{\text{total\_steps}}$ represent the fraction of training completion, where current_step is the current training iteration (e.g., batch or epoch, 0-indexed) and total_steps is the total number of such iterations for the run. The weight for the next-token prediction loss, $w_n$, is designed to smoothly increase, following the formula:

$$w_n(p) = 1 - \cos\left(\frac{\pi}{2} \cdot p\right)$$

This schedule ensures that the next-token prediction (NTP) objective becomes increasingly dominant as training nears completion, aligning with the goal for NTP to be the primary objective by the end of the run. Concurrently, the weight for the contrastive loss, $w_c$, is determined by a separate schedule:

$$w_c(p) = \cos\left(\frac{\pi}{2} \cdot p\right) \cdot \min\left(2p, 1\right) \cdot C$$

Here, $C$ represents a constant scaling factor (0.4 for Phi and Gemma3 and 1.6 for Pixtral). This formulation for $w_c$ incorporates two dynamic components based on the training progress $p$. A cosine decay term, $\cos\left(\frac{\pi}{2} \cdot p\right)$, which smoothly transitions the base weight from 1 down to 0. A linear ramp-up term, $\min\left(2 \cdot p, 1\right)$, which effectively scales the influence of the cosine term. This ramp-up is active during the first half of the training duration, remaining at 1 for the second half. The product of these terms, further scaled by $C$, allows the contrastive loss to have an initial warm-up period and then a gradually diminishing influence as the training increasingly prioritizes the NTP objective towards its conclusion. Both prompt tuning and LoRA training was conducted using dual cosine schedule. Progress for prompt tuning was calculated using epochs; for LoRA, progress was calculated using batches.

## J    LoRA ON VISION ENCODER

To further investigate the claim that the primary performance bottleneck resides within the LLM's reasoning pathways, we conducted an ablation study. We compare the final generative accuracy of our standard approach, where LoRA is applied to both the vision encoder and language model, against a configuration where same LoRA is applied only to the language model's attention layers. This analysis was performed across all three models, both datasets, and both training objectives to ensure the robustness of our findings.

Table A.4: Ablation of Vision Encoder LoRA. This table compares the final generative accuracy (%) when LoRA is applied to both the vision encoder and LLM versus only the LLM. The comparison is shown for both the standard next-token prediction ($\mathcal{L}_{NT}$) and the combined contrastive ($\mathcal{L}_{combined}$) objectives. The results show no statistically significant performance difference across these configurations.

| Model | Dataset | Method | Both (VE + LLM) | LLM-only |
|---|---|---|---|---|
| Phi | OpenWorld | Direct baseline | $59.0 \pm 4.3$ | $59.0 \pm 4.3$ |
| | | LoRA ($\mathcal{L}_{NT}$) | $92.2 \pm 2.4$ | $92.4 \pm 2.3$ |
| | | LoRA ($\mathcal{L}_{combined}$) | $95.6 \pm 1.8$ | $95.6 \pm 1.8$ |
| | HOI | Direct baseline | $52.1 \pm 3.5$ | $52.1 \pm 3.5$ |
| | | LoRA ($\mathcal{L}_{NT}$) | $78.6 \pm 2.8$ | $77.5 \pm 2.9$ |
| | | LoRA ($\mathcal{L}_{combined}$) | $79.2 \pm 2.8$ | $77.9 \pm 2.9$ |
| Pixtral | OpenWorld | Direct baseline | $72.4 \pm 3.9$ | $72.4 \pm 3.9$ |
| | | LoRA ($\mathcal{L}_{NT}$) | $93.4 \pm 2.2$ | $93.6 \pm 2.1$ |
| | | LoRA ($\mathcal{L}_{combined}$) | $95.0 \pm 1.9$ | $94.8 \pm 1.9$ |
| | HOI | Direct baseline | $57.8 \pm 3.4$ | $57.8 \pm 3.4$ |
| | | LoRA ($\mathcal{L}_{NT}$) | $78.0 \pm 2.9$ | $78.9 \pm 2.8$ |
| | | LoRA ($\mathcal{L}_{combined}$) | $79.6 \pm 2.8$ | $79.5 \pm 2.8$ |
| Gemma3 4B | OpenWorld | Direct baseline | $76.0 \pm 3.7$ | $76.0 \pm 3.7$ |
| | | LoRA ($\mathcal{L}_{NT}$) | $92.4 \pm 2.3$ | $92.4 \pm 2.3$ |
| | | LoRA ($\mathcal{L}_{combined}$) | $95.6 \pm 1.8$ | $95.6 \pm 1.8$ |
| | HOI | Direct baseline | $56.5 \pm 3.4$ | $56.5 \pm 3.4$ |
| | | LoRA ($\mathcal{L}_{NT}$) | $84.2 \pm 2.5$ | $84.2 \pm 2.5$ |
| | | LoRA ($\mathcal{L}_{combined}$) | $84.2 \pm 2.5$ | $84.2 \pm 2.5$ |

The results presented in Table A.4 are remarkably consistent. In every tested scenario, there is no statistically significant performance difference between applying LoRA adapters to the LLM alone versus applying them to both the vision encoder and the LLM.

This provides evidence that the LoRA adapters on the vision encoder are redundant for improving final task performance. It confirms that the gains from fine-tuning are not derived from adapting the initial feature extraction process, but rather from unlocking and refining the latent reasoning capabilities within the language model. This holds true even for the $\mathcal{L}_{combined}$ objective, demonstrating that co-adapting the vision encoder is unnecessary for the LLM to learn how to structure its final representations more effectively.

## K  PEFT COMPARISON

To understand how best to tackle the alignment gap for this task, we compared two primary PEFT methods: **prompt tuning**, which seeks to activate latent abilities by finding an optimal input, and **LoRA**, which adapts the model's core weights. We evaluated these methods using two distinct training objectives: the standard next-token prediction loss ($\mathcal{L}_{\text{NT}}$) and our combined objective ($\mathcal{L}_{\text{combined}}$), which adds a contrastive loss to explicitly improve representation quality. To further probe the limits of these methods, we tested both default and aggressive hyperparameter configurations for prompt-based tuning.

Our evaluation also measured structural robustness. The in-distribution (ID) test uses the standard interleaved prompt structure seen during training, while the out-of-distribution (OOD) test uses a labeled prompt structure that groups images by category, challenging the model's ability to generalize beyond the training syntax.

Table A.5: Comparison of PEFT methods. In the generative columns, **bold** indicates the LSC is surpassed. Superscripts on embedding classification scores indicate a statistically significant ($p < 0.05$) dependence between the item-level predictions of the similarity-based classifier and a generative method: I for the ID method and O for the OOD method. By default we use batch size of 25 and learning rate of $1e^{-4}$. Postfix tuning uses an aggressive batch size of 2 and learning rate of 0.01.

| Model | Dataset | Method | Generative (ID, %) | Generative (OOD, %) | LSC (%) | Final rep. (ID, %) | Final rep. (OOD, %) |
|---|---|---|---|---|---|---|---|
| Phi | OpenWorld | Direct baseline | $59.0 \pm 4.3$ | $79.4 \pm 3.5$ | $84.0 \pm 3.2^{I}$ | $76.4 \pm 3.7^{-I}$ | $78.2 \pm 3.6$ |
| | | Postfix tuning ($\mathcal{L}_{\text{NT}}$) | $\mathbf{94.2 \pm 2.0}$ | $\mathbf{90.2 \pm 2.6}$ | $84.2 \pm 3.2$ | $77.4 \pm 3.7^{I}$ | $78.8 \pm 3.6^{O}$ |
| | | Prompt tuning ($\mathcal{L}_{\text{NT}}$) | $\mathbf{90.4 \pm 2.6}$ | $83.4 \pm 3.3$ | $84.2 \pm 3.2$ | $74.6 \pm 3.8^{I}$ | $76.6 \pm 3.7^{O}$ |
| | | Prompt tuning ($\mathcal{L}_{\text{combined}}$) | $\mathbf{94.4 \pm 2.0}$ | $86.0 \pm 3.0$ | $84.2 \pm 3.2^{I}$ | $94.2 \pm 2.0^{I}$ | $93.2 \pm 2.2$ |
| | | LoRA ($\mathcal{L}_{\text{NT}}$) | $\mathbf{92.2 \pm 2.4}$ | $\mathbf{89.8 \pm 2.7}$ | $84.4 \pm 3.2^{I}$ | $74.8 \pm 3.8$ | $78.2 \pm 3.6^{O}$ |
| | | LoRA ($\mathcal{L}_{\text{combined}}$) | $\mathbf{95.6 \pm 1.8}$ | $\mathbf{94.4 \pm 2.0}$ | $84.4 \pm 3.2^{I}$ | $93.8 \pm 2.1^{I}$ | $91.2 \pm 2.5^{O}$ |
| | HOI | Direct baseline | $52.1 \pm 3.5$ | $58.8 \pm 3.4$ | $71.9 \pm 3.1$ | $60.5 \pm 3.4^{-I}$ | $60.6 \pm 3.4$ |
| | | Postfix tuning ($\mathcal{L}_{\text{NT}}$) | $63.2 \pm 3.3$ | $59.1 \pm 3.4$ | $71.9 \pm 3.1^{I,O}$ | $60.9 \pm 3.4^{I}$ | $61.8 \pm 3.4^{O}$ |
| | | Prompt tuning ($\mathcal{L}_{\text{NT}}$) | $58.5 \pm 3.4$ | $52.6 \pm 3.5$ | $71.9 \pm 3.1^{I}$ | $59.9 \pm 3.4^{I}$ | $61.9 \pm 3.4^{O}$ |
| | | Prompt tuning ($\mathcal{L}_{\text{combined}}$) | $65.4 \pm 3.3$ | $62.1 \pm 3.4$ | $71.9 \pm 3.1^{I}$ | $68.8 \pm 3.2^{I}$ | $62.9 \pm 3.3^{O}$ |
| | | LoRA ($\mathcal{L}_{\text{NT}}$) | $\mathbf{78.6 \pm 2.8}$ | $72.8 \pm 3.1$ | $71.9 \pm 3.1^{I,O}$ | $63.6 \pm 3.3^{I}$ | $64.4 \pm 3.3$ |
| | | LoRA ($\mathcal{L}_{\text{combined}}$) | $\mathbf{79.2 \pm 2.8}$ | $73.5 \pm 3.1$ | $71.9 \pm 3.1^{I,O}$ | $82.0 \pm 2.7^{I}$ | $82.9 \pm 2.6^{O}$ |
| Pixtral | OpenWorld | Direct baseline | $72.4 \pm 3.9$ | $79.4 \pm 3.5$ | $76.0 \pm 3.7^{I,O}$ | $87.2 \pm 2.9^{I}$ | $88.0 \pm 2.8^{O}$ |
| | | Postfix tuning ($\mathcal{L}_{\text{NT}}$) | $\mathbf{93.6 \pm 2.1}$ | $\mathbf{93.6 \pm 2.1}$ | $76.6 \pm 3.7$ | $87.6 \pm 2.9$ | $87.6 \pm 2.9$ |
| | | Prompt tuning ($\mathcal{L}_{\text{NT}}$) | $\mathbf{93.6 \pm 2.1}$ | $79.2 \pm 3.8$ | $76.6 \pm 3.7^{I}$ | $78.4 \pm 3.6$ | $83.0 \pm 3.3^{O}$ |
| | | Prompt tuning ($\mathcal{L}_{\text{combined}}$) | $\mathbf{94.4 \pm 2.0}$ | $\mathbf{88.2 \pm 2.8}$ | $76.6 \pm 3.7$ | $95.0 \pm 1.9^{I}$ | $93.4 \pm 2.2^{O}$ |
| | | LoRA ($\mathcal{L}_{\text{NT}}$) | $\mathbf{93.4 \pm 2.2}$ | $\mathbf{90.0 \pm 2.6}$ | $76.6 \pm 3.7$ | $87.2 \pm 2.9^{I}$ | $89.6 \pm 2.7$ |
| | | LoRA ($\mathcal{L}_{\text{combined}}$) | $\mathbf{95.0 \pm 1.9}$ | $\mathbf{94.6 \pm 2.0}$ | $74.4 \pm 3.8$ | $96.2 \pm 1.7^{I}$ | $95.4 \pm 1.8^{O}$ |
| | HOI | Direct baseline | $57.8 \pm 3.4$ | $60.6 \pm 3.4$ | $62.7 \pm 3.4^{I,O}$ | $70.2 \pm 3.2^{I}$ | $70.5 \pm 3.2$ |
| | | Postfix tuning ($\mathcal{L}_{\text{NT}}$) | $66.4 \pm 3.3$ | $66.4 \pm 3.3$ | $62.7 \pm 3.4^{I,O}$ | $71.0 \pm 3.1^{I}$ | $71.0 \pm 3.1^{O}$ |
| | | Prompt tuning ($\mathcal{L}_{\text{NT}}$) | $58.6 \pm 3.4$ | $54.0 \pm 3.5$ | $62.7 \pm 3.4^{I}$ | $62.6 \pm 3.4^{I}$ | $63.1 \pm 3.3^{-O}$ |
| | | Prompt tuning ($\mathcal{L}_{\text{combined}}$) | $\mathbf{70.8 \pm 3.2}$ | $58.9 \pm 3.4$ | $62.7 \pm 3.4^{I}$ | $73.5 \pm 3.1^{I}$ | $72.2 \pm 3.1^{O}$ |
| | | LoRA ($\mathcal{L}_{\text{NT}}$) | $\mathbf{78.0 \pm 2.9}$ | $\mathbf{74.4 \pm 3.0}$ | $61.6 \pm 3.4^{I,O}$ | $74.9 \pm 3.0^{I}$ | $73.2 \pm 3.1^{O}$ |
| | | LoRA ($\mathcal{L}_{\text{combined}}$) | $\mathbf{79.6 \pm 2.8}$ | $62.0 \pm 3.4$ | $63.1 \pm 3.3^{I}$ | $77.8 \pm 2.9^{I}$ | $77.6 \pm 2.9^{O}$ |
| Gemma3 4B | OpenWorld | Direct baseline | $76.0 \pm 3.7$ | $68.6 \pm 4.1$ | $89.8 \pm 2.7$ | $50.0^{I}$ | $50.0^{O}$ |
| | | Postfix tuning ($\mathcal{L}_{\text{NT}}$) | $86.0 \pm 3.0$ | $86.0 \pm 3.0$ | $89.8 \pm 2.7$ | $50.0$ | $50.0$ |
| | | Prompt tuning ($\mathcal{L}_{\text{NT}}$) | $86.0 \pm 3.0$ | $71.2 \pm 4.0$ | $89.8 \pm 2.7^{I}$ | $50.0$ | $50.2 \pm 4.4^{O}$ |
| | | Prompt tuning ($\mathcal{L}_{\text{combined}}$) | $82.2 \pm 3.4$ | $64.6 \pm 4.2$ | $89.8 \pm 2.7^{I}$ | $50.8 \pm 4.4^{-I}$ | $50.4 \pm 4.4^{O}$ |
| | | LoRA ($\mathcal{L}_{\text{NT}}$) | $92.4 \pm 2.3$ | $\mathbf{94.0 \pm 2.1}$ | $89.8 \pm 2.7^{I}$ | $50.0$ | $50.0$ |
| | | LoRA ($\mathcal{L}_{\text{combined}}$) | $\mathbf{95.6 \pm 1.8}$ | $\mathbf{96.2 \pm 1.7}$ | $89.8 \pm 2.7^{O}$ | $96.6 \pm 1.6^{I}$ | $96.8 \pm 1.5^{O}$ |
| | HOI | Direct baseline | $56.5 \pm 3.4$ | $54.2 \pm 3.5$ | $74.1 \pm 3.0$ | $50.0^{I}$ | $50.0^{O}$ |
| | | Postfix tuning ($\mathcal{L}_{\text{NT}}$) | $56.1 \pm 3.4$ | $56.1 \pm 3.4$ | $74.1 \pm 3.0$ | $50.0^{-I}$ | $50.0^{-O}$ |
| | | Prompt tuning ($\mathcal{L}_{\text{NT}}$) | $54.1 \pm 3.5$ | $51.6 \pm 3.5$ | $74.1 \pm 3.0^{O}$ | $50.0^{I}$ | $50.0^{-O}$ |
| | | Prompt tuning ($\mathcal{L}_{\text{combined}}$) | $58.9 \pm 3.4$ | $54.0 \pm 3.5$ | $74.1 \pm 3.0^{I}$ | $50.0^{-I}$ | $50.0^{O}$ |
| | | LoRA ($\mathcal{L}_{\text{NT}}$) | $\mathbf{84.2 \pm 2.5}$ | $\mathbf{81.8 \pm 2.7}$ | $74.1 \pm 3.0^{I,O}$ | $50.0$ | $50.0$ |
| | | LoRA ($\mathcal{L}_{\text{combined}}$) | $\mathbf{84.2 \pm 2.5}$ | $67.4 \pm 3.2$ | $74.1 \pm 3.0^{I}$ | $83.2 \pm 2.6^{I}$ | $82.1 \pm 2.7^{O}$ |

The results in Table A.5 reveal two key insights regarding the methods for resolving the alignment gap for this task.

**Mechanistic divergence towards a common goal.** A primary finding is that multiple strategies exist to achieve high generative accuracy. For instance, postfix tuning with aggressive hyperparameters can yield generative performance comparable to methods using the $\mathcal{L}_{\text{combined}}$ objective. However, this equivalence in outcome should not be mistaken for an equivalence in mechanism. The aggressive $\mathcal{L}_{\text{NT}}$ tuning likely discovers a more effective **non-linear decision logic**, enhancing the model's computational pathway without explicitly improving the linear structure of its representations. In contrast, the $\mathcal{L}_{\text{combined}}$ objective provides a more principled method for enhancing the **linear separability** of the final embeddings themselves. Thus, while both can surpass the LSC, they do so via different strategies: one by optimizing the downstream computation, the other by directly structuring the representations for that computation.

**A trade-off between representation quality and structural robustness.** The second key insight is a trade-off between representation quality and generalization to new prompt formats. Methods that are less invasive or only intervene late in the sequence, such as LoRA with $\mathcal{L}_{\text{NT}}$ and especially postfix tuning, demonstrate strong structural robustness, maintaining high performance on OOD prompts. Conversely, methods that either explicitly restructure representations ($\mathcal{L}_{\text{combined}}$) or introduce learnable tokens at the start of the sequence (standard prompt tuning) can exhibit brittleness. For example, on the HOI dataset, LoRA trained with $\mathcal{L}_{\text{combined}}$ shows a significant performance drop on the OOD prompt, despite its final representations remaining highly separable. This suggests that forcing a specific geometric structure can lead to syntactic overfitting, where the model's generative pathway learns to depend on the training prompt's structure, compromising its ability to reason flexibly.

## L RESULTS ACROSS HOI SPLITS

Table A.6 provides a detailed breakdown of model performance across the four distinct test splits of the Bongard-HOI dataset.

Table A.6: Performance on HOI - detailed breakdown

| Model | Method | Objective | Seen obj. Seen act. (%) | Seen obj. Unseen act. (%) | Unseen obj. Seen act. (%) | Unseen obj. Unseen act. (%) |
|---|---|---|---|---|---|---|
| Phi | Baseline gen. | | $51.5 \pm 6.9$ | $55.0 \pm 6.9$ | $51.0 \pm 6.9$ | $51.0 \pm 6.9$ |
| | Baseline sim. | | $61.5 \pm 6.7$ | $60.0 \pm 6.8$ | $57.5 \pm 6.9$ | $63.0 \pm 6.7$ |
| | Prompt tuning gen. | | $60.5 \pm 6.8$ | $59.0 \pm 6.8$ | $52.5 \pm 6.9$ | $62.0 \pm 6.7$ |
| | Prompt tuning sim. | | $61.5 \pm 6.7$ | $59.0 \pm 6.8$ | $57.0 \pm 6.9$ | $62.0 \pm 6.7$ |
| | Prompt tuning gen. | $\mathcal{L}_{\text{sim}}$ | $66.5 \pm 6.5$ | $69.0 \pm 6.4$ | $62.5 \pm 6.7$ | $63.5 \pm 6.7$ |
| | Prompt tuning sim. | $\mathcal{L}_{\text{sim}}$ | $63.5 \pm 6.7$ | $73.5 \pm 6.1$ | $70.5 \pm 6.3$ | $67.5 \pm 6.5$ |
| | LoRA gen. | $\mathcal{L}_{\text{NT}}$ | $79.5 \pm 5.6$ | $79.5 \pm 5.6$ | $75.5 \pm 6.0$ | $80.0 \pm 5.5$ |
| | LoRA sim. | $\mathcal{L}_{\text{NT}}$ | $65.0 \pm 6.6$ | $61.5 \pm 6.7$ | $61.5 \pm 6.7$ | $66.5 \pm 6.5$ |
| | LoRA gen. | $\mathcal{L}_{\text{combined}}$ | $77.0 \pm 5.8$ | $77.5 \pm 5.8$ | $81.0 \pm 5.4$ | $81.5 \pm 5.4$ |
| | LoRA sim. | $\mathcal{L}_{\text{combined}}$ | $82.0 \pm 5.3$ | $81.0 \pm 5.4$ | $77.5 \pm 5.8$ | $87.5 \pm 4.6$ |
| Pixtral | Baseline gen. | | $60.0 \pm 6.8$ | $61.5 \pm 6.7$ | $51.5 \pm 6.9$ | $58.0 \pm 6.8$ |
| | Baseline sim. | | $66.5 \pm 6.5$ | $73.5 \pm 6.1$ | $69.5 \pm 6.4$ | $71.5 \pm 6.3$ |
| | Prompt tuning gen. | | $60.5 \pm 6.8$ | $58.5 \pm 6.8$ | $56.5 \pm 6.9$ | $59.0 \pm 6.8$ |
| | Prompt tuning sim. | | $63.0 \pm 6.7$ | $62.0 \pm 6.7$ | $62.5 \pm 6.7$ | $63.0 \pm 6.7$ |
| | Prompt tuning gen. | $\mathcal{L}_{\text{sim}}$ | $73.0 \pm 6.2$ | $74.0 \pm 6.1$ | $64.0 \pm 6.7$ | $72.0 \pm 6.2$ |
| | Prompt tuning sim. | $\mathcal{L}_{\text{sim}}$ | $76.0 \pm 5.9$ | $73.5 \pm 6.1$ | $70.0 \pm 6.4$ | $74.5 \pm 6.0$ |
| | LoRA gen. | $\mathcal{L}_{\text{NT}}$ | $77.0 \pm 5.8$ | $79.5 \pm 5.6$ | $75.5 \pm 6.0$ | $80.0 \pm 5.5$ |
| | LoRA sim. | $\mathcal{L}_{\text{NT}}$ | $73.0 \pm 6.2$ | $77.5 \pm 5.8$ | $71.0 \pm 6.3$ | $78.0 \pm 5.7$ |
| | LoRA gen. | $\mathcal{L}_{\text{combined}}$ | $78.5 \pm 5.7$ | $81.5 \pm 5.4$ | $74.5 \pm 6.0$ | $84.0 \pm 5.1$ |
| | LoRA sim. | $\mathcal{L}_{\text{combined}}$ | $79.5 \pm 5.6$ | $78.5 \pm 5.7$ | $74.5 \pm 6.0$ | $78.5 \pm 5.7$ |
| Gemma3 4B | Baseline gen. | | $58.5 \pm 6.8$ | $56.0 \pm 6.9$ | $53.0 \pm 6.9$ | $58.5 \pm 6.8$ |
| | Baseline sim. | | $50.0$ | $50.0$ | $50.0$ | $50.0$ |
| | Prompt tuning gen. | | $53.0 \pm 6.9$ | $56.5 \pm 6.9$ | $53.0 \pm 6.9$ | $54.0 \pm 6.9$ |
| | Prompt tuning sim. | | $50.0$ | $50.0$ | $50.0$ | $50.0$ |
| | Prompt tuning gen. | $\mathcal{L}_{\text{sim}}$ | $56.5 \pm 6.9$ | $61.5 \pm 6.7$ | $59.5 \pm 6.8$ | $58.0 \pm 6.8$ |
| | Prompt tuning sim. | $\mathcal{L}_{\text{sim}}$ | $50.0$ | $50.0$ | $50.0$ | $50.0$ |
| | LoRA gen. | $\mathcal{L}_{\text{NT}}$ | $83.5 \pm 5.1$ | $84.5 \pm 5.0$ | $85.0 \pm 4.9$ | $84.0 \pm 5.1$ |
| | LoRA sim. | $\mathcal{L}_{\text{NT}}$ | $50.0$ | $50.0$ | $50.0$ | $50.0$ |
| | LoRA gen. | $\mathcal{L}_{\text{combined}}$ | $83.0 \pm 5.2$ | $84.0 \pm 5.1$ | $85.0 \pm 4.9$ | $85.0 \pm 4.9$ |
| | LoRA sim. | $\mathcal{L}_{\text{combined}}$ | $84.0 \pm 5.1$ | $86.5 \pm 4.7$ | $82.5 \pm 5.3$ | $80.0 \pm 5.5$ |

There is no significant accuracy degradation when models are tested on unseen objects, unseen actions, or both simultaneously. In fact, for both Phi and Pixtral, the highest generative accuracy with the combined loss is achieved on the most difficult split (unseen obj. / unseen act.), providing strong evidence that the models are learning the abstract relational concept rather than memorizing training examples.

## M  DOMAIN GENERALIZATION

To rigorously assess whether the reasoning skills learned during fine-tuning are truly generalizable, we conducted two distinct cross-domain evaluations. The first tests for generalization across different Bongard-style tasks (OpenWorld and HOI). The second, more challenging evaluation tests whether the learned skills transfer to a task with a completely different structure: the text-image retrieval task of the Winoground dataset.

### M.1  CROSS-BONGARD GENERALIZATION

In this evaluation, models were fine-tuned on one Bongard dataset (e.g., OpenWorld) and evaluated on the other (e.g., HOI) to assess their ability to generalize beyond the training domain's specific concepts and structures. The dataset column in Table A.7 indicates the evaluation dataset (i.e., the model was trained on the other dataset).

Table A.7: Cross-Bongard generalization performance. Models were trained on one dataset and evaluated on the other. **bold** indicates surpassing of LSC. Superscripts on similarity scores denote a significant dependence ($p < 0.05$) with the predictions of the corresponding generative method (G).

| Model | Dataset | Method | Generative (OOD, %) | LSC (%) | Final rep. (OOD, %) |
|---|---|---|---|---|---|
| Phi | OpenWorld | Direct baseline | $79.4 \pm 3.5$ | $84.0 \pm 3.2$ | $78.2 \pm 3.6$ |
| | | LoRA (HOI, $\mathcal{L}_{\text{NT}}$) | $\mathbf{90.0 \pm 2.6}$ | $84.4 \pm 3.2^G$ | $78.4 \pm 3.6^G$ |
| | | LoRA (HOI, $\mathcal{L}_{\text{combined}}$) | $\mathbf{92.8 \pm 2.3}$ | $84.4 \pm 3.2^G$ | $89.2 \pm 2.7^G$ |
| | HOI | Direct baseline | $58.8 \pm 3.4$ | $71.9 \pm 3.1$ | $60.6 \pm 3.4$ |
| | | LoRA (OpenWorld, $\mathcal{L}_{\text{NT}}$) | $64.6 \pm 3.3$ | $71.9 \pm 3.1^G$ | $60.6 \pm 3.4$ |
| | | LoRA (OpenWorld, $\mathcal{L}_{\text{combined}}$) | $67.4 \pm 3.2$ | $71.9 \pm 3.1^G$ | $67.2 \pm 3.3^G$ |
| Pixtral | OpenWorld | Direct baseline | $79.4 \pm 3.5$ | $76.0 \pm 3.7^G$ | $88.0 \pm 2.8^G$ |
| | | LoRA (HOI, $\mathcal{L}_{\text{NT}}$) | $\mathbf{88.8 \pm 2.8}$ | $76.6 \pm 3.7^G$ | $89.0 \pm 2.7^G$ |
| | | LoRA (HOI, $\mathcal{L}_{\text{combined}}$) | $\mathbf{83.4 \pm 3.3}$ | $74.4 \pm 3.8^G$ | $87.4 \pm 2.9^G$ |
| | HOI | Direct baseline | $60.6 \pm 3.4$ | $62.7 \pm 3.4^G$ | $70.5 \pm 3.2$ |
| | | LoRA (OpenWorld, $\mathcal{L}_{\text{NT}}$) | $64.9 \pm 3.3$ | $61.6 \pm 3.4^G$ | $72.5 \pm 3.1^G$ |
| | | LoRA (OpenWorld, $\mathcal{L}_{\text{combined}}$) | $\mathbf{71.0 \pm 3.1}$ | $63.1 \pm 3.3^G$ | $68.4 \pm 3.2^G$ |
| Gemma3 4B | OpenWorld | Direct baseline | $68.6 \pm 4.1$ | $89.8 \pm 2.7$ | $50.0^G$ |
| | | LoRA (HOI, $\mathcal{L}_{\text{NT}}$) | $87.2 \pm 2.9$ | $89.8 \pm 2.7^G$ | $50.0$ |
| | | LoRA (HOI, $\mathcal{L}_{\text{combined}}$) | $83.4 \pm 3.3$ | $89.8 \pm 2.7$ | $83.2 \pm 3.3^G$ |
| | HOI | Direct baseline | $54.2 \pm 3.5$ | $74.1 \pm 3.0$ | $50.0^G$ |
| | | LoRA (OpenWorld, $\mathcal{L}_{\text{NT}}$) | $69.0 \pm 3.2$ | $74.1 \pm 3.0^G$ | $50.0$ |
| | | LoRA (OpenWorld, $\mathcal{L}_{\text{combined}}$) | $65.8 \pm 3.3$ | $74.1 \pm 3.0^G$ | $67.4 \pm 3.2^G$ |

The results indicate that models can generalize their reasoning strategies to novel concepts within the same task paradigm. Pixtral demonstrates the most robust generalization, surpassing its LSC in both transfer directions. Phi shows strong generalization when trained on the broader relational concepts of HOI and tested on OpenWorld, but this success is not bidirectional. Across all models, a key observation is that generative accuracy remains tightly coupled with the final representation classification accuracy when trained with the combined loss ($\mathcal{L}_{\text{combined}}$), suggesting the learned alignment is a general property, not specific to the training data distribution

## M.2 GENERALIZATION TO WINOGROUND

To test if fine-tuning imparts a more fundamental reasoning capability, we evaluated the models on the Winoground benchmark without task-specific adaptation. This benchmark provides a compelling test case, as its text-retrieval component directly evaluates the inter-image comparison skills honed during our training. The results in Table A.8 reveal how the nature of the training task and the fine-tuning objective dictates the success of cross-task generalization.

Table A.8: Zero-shot cross-task generalization to the Winoground dataset. The CLIP baseline is a linear probe on the vision encoder of the Phi model (and its respective text encoder), serving as an LSC-like benchmark for that model.

| Model | Train dataset | Objective | Text retrieval acc. (%) | Image retrieval acc. (%) |
|---|---|---|---|---|
| Phi | Baseline | | 16.75 | 2.50 |
| | CLIP Baseline (ViT-L/14) | | 27.75 | 11.75 |
| | OpenWorld | $\mathcal{L}_{NT}$ | 24.00 | 14.75 |
| | | $\mathcal{L}_{combined}$ | 18.75 | 1.75 |
| | HOI | $\mathcal{L}_{NT}$ | 18.25 | 13.50 |
| | | $\mathcal{L}_{combined}$ | **29.25** | **14.00** |
| Pixtral | Baseline | | 28.75 | 4.25 |
| | OpenWorld | $\mathcal{L}_{NT}$ | 32.75 | 19.75 |
| | | $\mathcal{L}_{combined}$ | 33.25 | 8.50 |
| | HOI | $\mathcal{L}_{NT}$ | 43.50 | **24.75** |
| | | $\mathcal{L}_{combined}$ | **54.75** | 12.25 |
| Gemma3 4B | Baseline | | 5.25 | 0.75 |
| | OpenWorld | $\mathcal{L}_{NT}$ | 5.75 | 1.00 |
| | | $\mathcal{L}_{combined}$ | 12.25 | 2.75 |
| | HOI | $\mathcal{L}_{NT}$ | 5.75 | 3.25 |
| | | $\mathcal{L}_{combined}$ | **12.25** | **13.25** |

**Task-specific skill transfer.** Generalization is highly dependent on the alignment between the skills cultivated during training and those required by the evaluation task. Fine-tuning on OpenWorld, which focuses on atomic semantic concepts, resulted in poor transfer to the compositional reasoning required by Winoground. This mirrors our findings in the cross-Bongard evaluation, suggesting that the model learned a task-specific strategy that was not broadly applicable to compositional challenges.

**Objective-driven alignment.** Training on the relational HOI dataset proved far more effective for transferring to Winoground. Within this setting, the $\mathcal{L}_{combined}$ objective consistently improved text retrieval accuracy over the standard $\mathcal{L}_{NT}$ loss across all models. This is most pronounced for Pixtral, whose accuracy jumps from 43.50% to 54.75%. This demonstrates that our combined objective successfully aligns the model's structured visual representations and its generative pathway.

**Re-emergence of the LSC.** The Winoground evaluation of the Phi model offers a compelling illustration of our framework in a new domain. The untuned model's generative performance (16.75%) exhibits an alignment gap, falling well below the LSC (27.75%) established by a linear probe on its vision and respective text encoder. Fine-tuning on the relational HOI task with our $\mathcal{L}_{combined}$ objective closes this deficit, elevating the end-to-end performance to 29.25%. However, this improvement is not statistically significant over the LSC. Consequently, by our framework's definition, an alignment gap persists. This indicates that the model has learned to exploit the linearly decodable information within its representations but has not yet developed a sophisticated (non-linear) reasoning capability that provides a performance advantage beyond this ceiling on a new task. We hypothesize that achieving such an improvement would require fine-tuning on a dataset that more comprehensively covers the specific compositional and relational concepts tested by the Winoground benchmark.

# N    TRAINING SENSITIVITY TO AUXILIARY CONTRASTIVE OBJECTIVE

To demonstrate the impact of loss ($\mathcal{L}_{combined} = w_n\mathcal{L}_{NT} + w_c\mathcal{L}_{sim}$) counterparts, we conducted a hyperparameter sweep on the Phi-3.5 model using the HOI dataset on a fixed seed. We compare two scheduling strategies: a **constant** schedule, where $w_c$ remains fixed throughout training and $w_n$ is fixed to 1, and a **dual cosine** schedule (detailed in Appendix I), designed to provide a strong initial structural signal that gradually yields to the next-token prediction objective. Results are summarized in Table A.9.

Table A.9: Sensitivity analysis of $\mathcal{L}_{combined}$ hyperparameters on Phi-3.5 (HOI dataset). We compare the constant schedule against dual cosine schedule. The CLIP baseline is a linear probe on the vision encoder of the Phi model (and its respective text encoder), serving as an LSC-like benchmark for that model. **Final rep.** denotes the linear separability of final representations, and **Wino.** denotes zero-shot text and image retrieval accuracies on Winoground. The in-distribution (ID) test uses interleaved prompt structure seen during training, while the out-of-distribution (OOD) test uses a labeled prompt structure.

| Method | Schedule | Generative (ID, %) | Generative (OOD, %) | Final rep. (ID, %) | Final rep. (OOD, %) | Wino. text (%) | Wino. image (%) |
|---|---|---|---|---|---|---|---|
| Direct baseline | — | 52.1 | 58.8 | 60.5 | 60.6 | 16.75 | 2.50 |
| CLIP baseline (ViT-L/14) | — | — | — | 71.9 | 71.9 | 27.75 | 11.75 |
| $w_c = 0.0$ ($\mathcal{L}_{NT}$) | constant | 76.00 | 68.63 | 61.63 | 63.38 | 13.00 | 13.50 |
| $w_c = 0.025$ | constant | 75.63 | 69.00 | 62.88 | 64.00 | 12.00 | 13.50 |
| $w_c = 0.05$ | constant | 76.88 | 70.13 | 67.25 | 67.75 | 13.00 | 11.00 |
| $w_c = 0.1$ | constant | **77.50** | 70.50 | 71.75 | 74.63 | 12.00 | 11.50 |
| $w_c = 0.2$ | constant | 77.00 | **71.13** | 76.00 | 76.50 | 14.00 | 11.00 |
| $w_c = 0.4$ | constant | 76.13 | 70.50 | 81.00 | 80.88 | 14.00 | 10.75 |
| $w_c = 0.8$ | constant | 74.00 | 62.38 | 76.63 | 77.88 | 9.75 | 6.00 |
| $w_c = 1.6$ | constant | 74.38 | 66.63 | **82.25** | **82.13** | 21.75 | 14.00 |
| $w_c = 0.025$ | cosine | 76.88 | 70.75 | 81.50 | 79.50 | 17.25 | 11.50 |
| $w_c = 0.05$ | cosine | 74.25 | 64.13 | 81.63 | 80.63 | 21.00 | 11.75 |
| $w_c = 0.1$ | cosine | 76.88 | **72.13** | 82.00 | 81.75 | 22.00 | 13.50 |
| $w_c = 0.2$ | cosine | 74.00 | 67.13 | 83.00 | 82.00 | **25.25** | 11.25 |
| $w_c = 0.4$ | cosine | **77.75** | 70.38 | 82.00 | 82.00 | 22.50 | 17.25 |
| $w_c = 0.8$ | cosine | 75.63 | 67.25 | 82.75 | 81.63 | 21.75 | 14.75 |
| $w_c = 1.6$ | cosine | 68.88 | 66.00 | **83.63** | **82.50** | 18.25 | 9.75 |

**Generative vs. representational alignment.**    Increasing $w_c$ improves the linear separability of the final representations. Under the constant schedule, probe accuracy rises monotonically from $61.63\%$ at $w_c = 0.0$ to $82.25\%$ at $w_c = 1.6$. However, generative accuracy peaks early at $w_c = 0.1$ ($77.50\%$) and degrades at higher values ($74.38\%$ at $w_c = 1.6$), indicating that excessive contrastive pressure can disrupt the model's generative capabilities.

**Impact of loss counterpart scheduling on generalization.**    The dual cosine schedule demonstrates superior efficiency in balancing these objectives. Even at low weights ($w_c = 0.025$), the cosine schedule achieves high probe accuracy ($81.50\%$) comparable to aggressive constant schedules, while maintaining high generative performance.

## O    EXAMPLE SEMANTIC CONCEPTS

The Bongard OpenWorld dataset primarily utilizes semantic concepts based on objects, scenes, actions, or attributes:
- Elderly person using a cell phone.
- Evening sunset on the desert dunes.
- People playing water polo in the swimming pool.
- People are in a hurry.
- A boat tied with a rope in the water.
- A player shooting on the hockey field.
- A cute dog wearing a cozy sweater.
- Solar panels on the house roof.
- The shepherd herds flocks of sheep.
- A closeup of a dandelion.
- Vehicle tires (e.g., steel frame for car tires, motorcycle tires).
- Chimney on the house roof.

The HOI dataset utilizes semantic concepts based on human-object interactions:
- A person carrying a surfboard.
- A person riding a surfboard.
- A person holding an apple.
- A person swinging a tennis racket.
- Multiple people sitting on a bench.
- A person brushing with a toothbrush.
- A person adjusting or tying a tie.
- A person using a mouse.
- A person throwing a frisbee.
- A person holding and about to eat an apple.
- A person peeling or cutting an apple.
- A person lying on a bench.

# P VISUALIZING THE MECHANISM OF PEFT VIA ATTENTION MAPS

The quantitative results in the main paper demonstrate that PEFT can effectively bridge and surpass the LSC. To understand *how* these interventions alter the model's internal computations, this appendix provides a qualitative, mechanistic explanation by visualizing the models' attention patterns before and after fine-tuning.

Attention maps reveal the core mechanism of the transformer architecture, showing which parts of the input sequence (text and image tokens) are attended to when building updated representations. A higher attention score (brighter color) indicates a stronger influence. By comparing these maps, we can directly observe how fine-tuning reshapes the information flow within the model to solve the reasoning task, providing visual evidence that complements the quantitative analysis.

## P.1 DISTINCT ARCHITECTURAL SIGNATURES AND TUNING MECHANISMS

Our analysis reveals that each model family possesses a unique baseline attention strategy, which dictates the mechanism of improvement unlocked by fine-tuning.

**Gemma3: overcoming representational degradation.** As established in the main paper, Gemma3 models exhibit a high LSC but suffer from severe representational degradation, where the final-layer embeddings become poorly separated. The attention maps in Figure A.4 reveal the architectural reason and the solution. Gemma3 uses an efficient **sliding window attention** for the most part (narrow diagonal band) (layer 24 behaved differently for some reason).

- With $\mathcal{L}_{\text{NT}}$ tuning, the model learns the task but the attention mechanism is not visibly altered; it refines computations within its existing pathways.

- With $\mathcal{L}_{\text{combined}}$ tuning, the contrastive loss induces a change that makes the model's reasoning process visible. In the final layers, the attention map develops more structured, bright vertical stripes. Each well-defined stripe represents a targeted "read" operation, where the model globally accesses the compressed representation of visual features. This structured, cross-image comparison provides a clear mechanistic explanation for how the model overcomes its baseline shortcoming. The explicit contrastive pressure forces it to develop and execute this comparison strategy, repairing its degraded final representations and aligning with the recovery of final layer linear separability, as seen in Table 5.

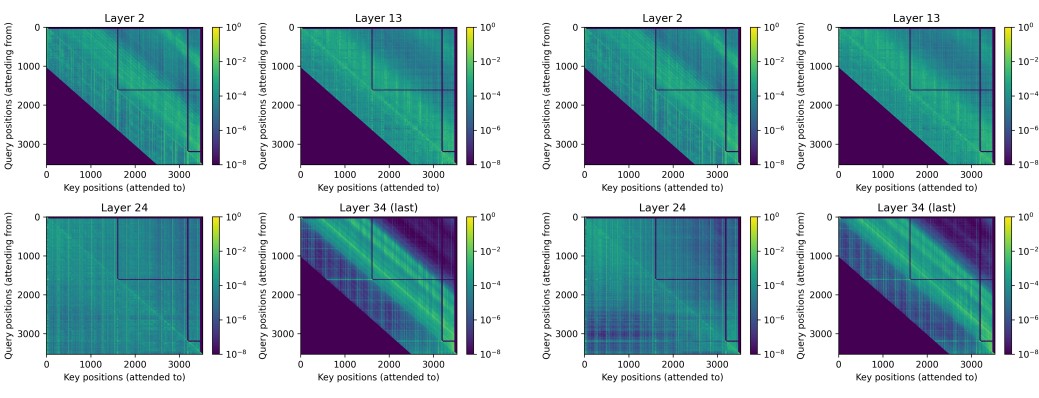

(a) Gemma3 4B after LoRA tuning ($\mathcal{L}_{\text{NT}}$)    (b) Gemma3 4B after LoRA tuning ($\mathcal{L}_{\text{combined}}$)

Figure A.4: Attention maps for Gemma3 4B. The contrastive loss in (b) visibly forces global cross-image attention in the final layers, a pattern not as pronounced in (a). This visualizes the mechanism that improves the model's internal representations. The vertical axis represents the token attending from and the horizontal axis represents the token attending to.

**Phi and Pixtral: representation refinement and feature enhancement.** Phi and Pixtral employ a standard causal attention mask across all layers (Figures A.5 and A.6). Their path to success is not about fixing a shortcoming but refining existing processes.

With Phi possessing a higher-quality LSC from the outset, Phi's fine-tuning is primarily a process of **refinement and noise reduction**. The attention maps visualize this as the patterns *within* each image block becoming more structured and less diffuse. Instead of a scattered focus across all visual patches, the model learns to consistently attend to the most salient features while ignoring irrelevant ones. This refined intra-image feature extraction allows for a cleaner signal to be passed to the final layers, enabling the model to better leverage its already-strong representations and surpass its LSC.

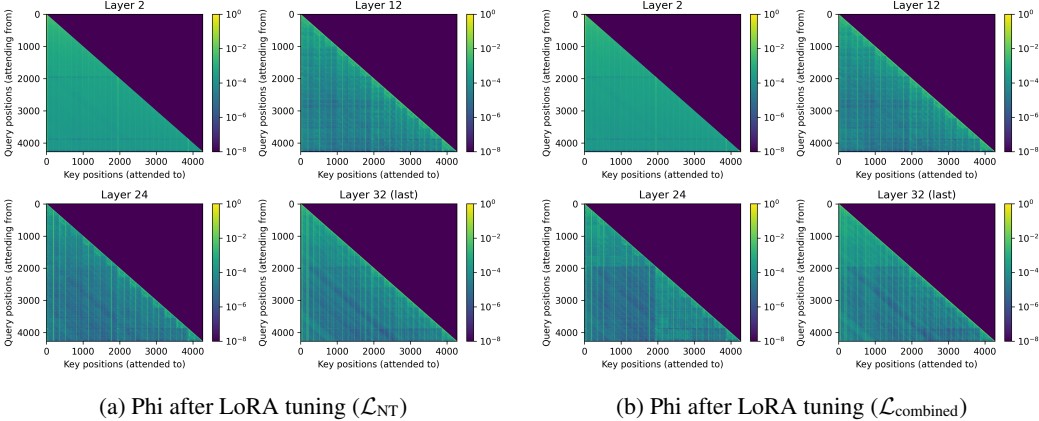

(a) Phi after LoRA tuning ($\mathcal{L}_{\text{NT}}$)      (b) Phi after LoRA tuning ($\mathcal{L}_{\text{combined}}$)

Figure A.5: Attention maps for the Phi model after LoRA tuning with (a) the $\mathcal{L}_{\text{NT}}$ objective and (b) the $\mathcal{L}_{\text{combined}}$ objective. Both training methods result in a similar outcome: a refinement of attention over the model's already high-quality initial representations. The patterns become visibly more structured and less diffuse, visualizing a mechanism of noise reduction and more targeted feature aggregation that enables the model to surpass its LSC. The vertical axis represents the token attending from and the horizontal axis represents the token attending to.

Pixtral's success, even at baseline, stems from its inherent ability to perform vision **representation refinement**. The attention maps reveal the mechanism: an intensified focus on intra-image processing in later layers, visible as more structured diagonal blocks. Crucially, this visual signature is nearly identical when fine-tuning with either the standard $\mathcal{L}_{\text{NT}}$ or the $\mathcal{L}_{\text{combined}}$ objective. This equivalence demonstrates that standard next-token prediction is sufficient to fully engage Pixtral's innate pathway for **extracting and enhancing salient features**.

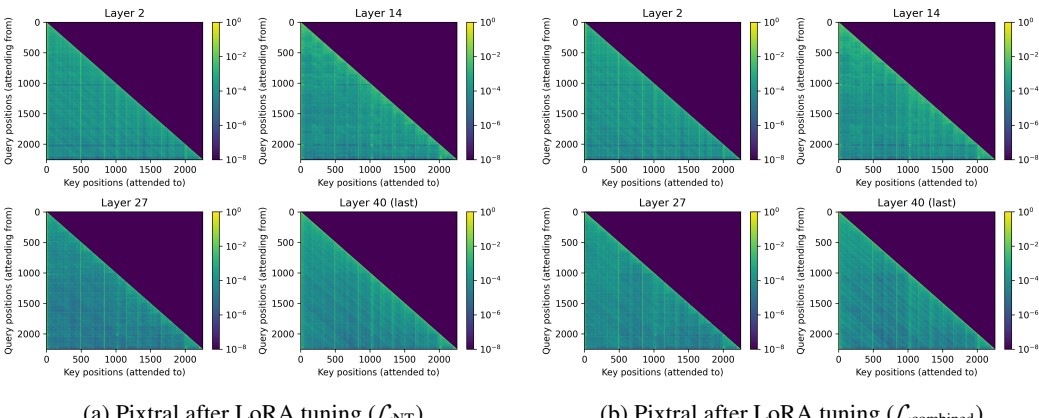

(a) Pixtral after LoRA tuning ($\mathcal{L}_{\text{NT}}$)        (b) Pixtral after LoRA tuning ($\mathcal{L}_{\text{combined}}$)

Figure A.6: Attention maps for the Pixtral model after LoRA tuning with (a) the $\mathcal{L}_{\text{NT}}$ objective and (b) the $\mathcal{L}_{\text{combined}}$ objective. The attention patterns are nearly identical, showing that both objectives activate the same underlying mechanism. The vertical axis represents the token attending from and the horizontal axis represents the token attending to.

