# OpenReview forum: "Beyond the Linear Separability Ceiling: Aligning Representations in VLMs"
_ICLR.cc/2026/Conference — Submitted to ICLR 2026_

### Official Review · Reviewer_jDo8 · 2025-10-26

**Soundness:** 2
**Presentation:** 2
**Contribution:** 2
**Rating:** 4
**Confidence:** 3

**Summary:**

The paper investigates why VLMs fail on abstract visual reasoning tasks such as the Bongard problems. The paper proposes the Linear Separability Ceiling (LSC), a diagnostic measure of how well a simple linear classifier performs on a VLM’s visual embeddings. It serves as a baseline to test whether a model’s reasoning pipeline adds non-linear reasoning beyond what its visual features already provide.
The authors evaluate several vLMS (e.g. Gemma, Pixtral, Phi, Qwen, InternVL) and realize that most models fail to exceed their own LSC, implying that their reasoning components are not effectively leveraging visual representations.
They also identify two strategies by which the VLMs can surpass the LSC
1) Enhanced linear separability: improving internal representations to be more discriminative
2) Non-linear decision logic: leveraging deeper reasoning pathways beyond linear readouts

The paper also proposes a finetuning method that combines next-token prediction and a contrastive loss to improve representation alignment. They show that this object allows models to surpass LSC and improve in-domain and cross-domain reasoning.

**Strengths:**

1) The concept of an “alignment gap” between perception and reasoning reframes common VLM failures through a geometric and statistical lens
2) The paper’s decomposition of reasoning into linear vs. non-linear computational mechanisms (perceptual refinement vs. reasoning) provides conceptual clarity that could generalize beyond VLMs

**Weaknesses:**

1) The experimental validation is limited to binary image-to-text retrieval variants of Bongard-style tasks (Bongard OpenWorld and Bongard HOI), where the model must choose between two options given a query image. While these are well-defined tests for abstract reasoning, they represent a highly specific and simplified evaluation setup. It remains unclear whether the proposed framework and fine-tuning method would generalize to broader vision-language reasoning tasks such as open-ended visual question answering, caption generation. The improvements demonstrated may partially stem from optimizing for the same binary discriminative signal introduced by the contrastive objective, rather than from a genuinely general enhancement of multimodal reasoning.
2) Because the proposed contrastive objective directly optimizes for improved separability between positive and negative examples, part of the observed performance gain could be attributed to alignment with the evaluation metric itself (linear separability), rather than improved reasoning. Additional experiments on independent tasks not directly linked to the contrastive loss would help verify generalization.
3) The paper notes that the combined objective (L_combined) can induce catastrophic forgetting and prompt-format overfitting. It is unclear whether the performance of the finetuned model on general vision-language benchmarks (e.g. VQA benchmarks) is preserved after finetuning.
Furthermore, the paper provides little quantitative analysis of how the weighting between the next-token prediction and contrastive loss terms affects this trade-off. A sensitivity study or learning dynamics analysis would strengthen the claims.

**Questions:**

1) The paper mentions LSC could be used as a “live diagnostic” during model training. Could the authors outline how such an online LSC metric might be integrated into training pipelines (e.g., as a stopping criterion or auxiliary signal)?
2) How sensitive are the results to the relative weights w_m and w_c in L_combined? Does a small contrastive component already improve alignment, or is a strong contrastive signal necessary?
3) Have the authors evaluated whether the LSC framework predicts reasoning performance on tasks that are less abstract (e.g., visual question answering or commonsense reasoning)? How consistent is the alignment gap in these settings?

---

> ### Author Response · Authors · 2025-11-17
>
> We would like to thank the reviewer for taking the time and going through the paper and suggesting ways to rephrase contributions to be more accurate.
>
> The weaknesses you pointed out are grounded and valid.
> We narrowed down the extent of our claims to image-to-image classification in the abstract and conclusion on page 9. We rephrased catastrophic forgetting to image-to-text misalignment as it’s more accurate based on the LSC framework. We discuss the implications and limitations in more detail in section 7 on page 9 and find that expansion of the framework to VQA could be a promising next direction, a direction we briefly explored in the new Appendix A on pages 15 to 17. There, we tested the generalization of our probing and training method on the VQA datasets.  TL;DR is we saw no statistically significant difference for when w_c <= 0.1.
> When we fine-tuned a language model to be a classifier using fixed prompt, then generalization to other tasks degraded regardless of objective.
>
> Answers to the questions:
> 1. When models are subjected to instruction-tuning or reinforcement learning, the task-specific learning signal can cause representational degradation, where linearly decodable information is diminished. The LSC framework can be used to detect such degradation. More context in the discussion section on page 9.
> 2. A sensitivity study is in Appendix A on page 16.
> 3. LCS on various VQA datasets for Phi is in Appendix A on page 16.

---

> ### Author Response · Authors · 2025-11-26
> **Follow-up**
>
> We have further updated the manuscript to provide the quantitative analyses you requested regarding sensitivity and generalization.
>
> To address your question on the sensitivity of the combined loss weights ($w_c$ and $w_n$), we added a detailed hyperparameter sweep in Appendix N. This analysis explicitly maps the trade-off between representational quality and generative capability, confirming the necessity of the dual cosine schedule to effectively balance these conflicting signals.
>
> Regarding the application of the LSC framework to less abstract tasks, we updated Appendix A to include LSC baselines for multiple-choice VQA tasks (HOI, POPE, A-OKVQA, and ScienceQA). These results demonstrate that the framework itself is useful beyond Bongard tasks.
>
> To address the concern that performance gains might stem solely from alignment with the evaluation metric, we added Appendix B. The Isomap visualizations reveal that our method promotes globally consistent 1D linear structures ("rays"), providing geometric evidence that the improvement is a fundamental structural refinement.
>
> We have also expanded on the limitations of "image-to-text" alignment gaps in the new "Discussion, limitations and future-work" section right before conclusion, and updated the contributions to be more concise. The new section should address your concerns in full detail.
>
> Do these additions, combined with the previous responses, sufficiently address your concerns regarding the robustness and scope of our work? Do you have any other concerns or questions?

---

### Official Review · Reviewer_3cSn · 2025-10-27

**Soundness:** 3
**Presentation:** 3
**Contribution:** 2
**Rating:** 4
**Confidence:** 4

**Summary:**

This paper investigates failures of VLMs on abstract visual reasoning tasks like Bongard problems, questioning whether the bottleneck lies in visual perception or higher-level reasoning. The authors introduce a diagnostic framework centered on the Linear Separability Ceiling (LSC): the maximum performance achievable by a linear classifier on the VLM's initial visual embeddings. Applying this framework, they discover an alignment gap: most state-of-the-art VLMs fail to generatively outperform their own LSC, suggesting their reasoning capabilities are poorly aligned with their visual representations. They propose a fine-tuning method using LoRA with a combined objective, adding a contrastive loss to the standard next-token prediction loss. This method improves the linear separability of final embeddings, successfully allowing models to systematically surpass the LSC and achieve higher performance on abstract reasoning tasks.

**Strengths:**

1. The graphs and tables are clear and easy to understand.
2. Experiments are thorough, covering multiple VLMs, datasets, PEFT methods, objectives, and generalization scenarios.

**Weaknesses:**

1. While effective, the LSC relies solely on linear separability. It's possible that representations hold complex non-linear structures useful for reasoning that the LSC metric fails to capture.
2. The core observation that VLM generative performance often fails to surpass a linear probe on its visual features is not entirely new. Similar gaps between representation quality and end-to-end performance have been previously studied, showing VLMs can underperform linear probes on classification or generally overlook information in their visual representations.
Some related works like:
[Why are Visually-Grounded Language Models Bad at Image Classification?](https://arxiv.org/pdf/2405.18415.pdf)
[Hidden in plain sight: VLMs overlook their visual representations](https://arxiv.org/pdf/2506.08008.pdf)

**Questions:**

1. Can the fine-tuning approach be successfully applied to improve VLM performance on other challenging reasoning domains beyond Bongard problems, such as VQA or complex instruction following?

---

> ### Author Response · Authors · 2025-11-17
>
> We thank the reviewer for taking the time to go through the paper and finding relevant papers.
>
> We will first address the weaknesses:
> 1. For initial visual features it’s possible, but the contrastive pretraining goal of vision encoder was to make them linearly separable.
> 2. We’ll go through the relevant papers’ contributions and expand how our work adds additional value.
>
> [Why are Visually-Grounded Language Models Bad at Image Classification?](https://arxiv.org/pdf/2405.18415)
>
> A1: They uncover a gap unaddressed by previous research.
>
> B1: We expand on this by showing that prompt format plays a big role here.
>
> A2: They find that lack of alignment data, rather than training objective, is the cause.
>
> B2: Alignment data certainly helps, but we provide a new perspective to the alignment gap. And what goes for the training objective, then it does make a difference.
>
> [Hidden in plain sight: VLMs overlook their visual representations](https://arxiv.org/pdf/2506.08008)
>
> A1: They find that shifting from standard visual probing strategies to a VLM-based evaluation results in a universal drop in performance. Additionally, vision representations throughout projector and LLM layers do not degrade.
>
> B1: Our findings indicate that it's more nuanced. With proper prompting the models can work surprisingly well, but with suboptimal prompting the performance does degrade. It ended up being an activation issue in the end, caused by alignment gap, where we provide a new perspective. What goes for visual representations themselves, we found that there are two types of approaches: representational refinement, where representations improve for the task at hand; and non-linear decision logic.
>
> A2: Prompt(prefix)-tuning the VLM improves performance marginally.
>
> B2: We discovered that when the learnable prompt is postfixed after the query with a meta-learning objective then performance increases significantly for cases when the initial issue was an activation issue.
>
> A3: Ability to use its vision representations is a limiting factor in VLM performance…(later in the text) we stand with prior work highlighting the importance of developing stronger vision encoders…
>
> B3: Yes, but we found that perception doesn’t really end. It’s a continuous process that heavily depends on the context and world knowledge. And to that end we demonstrate that the LLM part of the VLM can continue building the visual representations to achieve downstream results at least equal if not better than standard fine-tuning on abstract visual task.
>
> What goes for the question, then our initial implementation to VQA style task successfully aligned visual representations but no statistically significant downstream accuracy gains were noted. There are a lot of variables to consider, further detailed in the short and long conclusion in appendix A on page 17.

---

> ### Author Response · Authors · 2025-11-26
> **Follow-up**
>
> We have further updated the manuscript to provide deeper support regarding your concerns on the linearity assumption and the framework's applicability to broader domains.
>
> To address your concern that representations might hold complex non-linear structures missed by LSC, we added Isomap visualizations (Appendix B). These reveal that the model's intrinsic geometry organizes into 1D linear "rays", providing geometric evidence that linear separability is a fundamental property of these representations, thereby validating the LSC metric.
>
> Addressing your question on applying the framework to other domains, we expanded the LSC analysis to POPE, A-OKVQA, and ScienceQA (Appendix A). We established LSC baselines for these tasks, demonstrating that the diagnostic framework successfully identifies alignment gaps (if any) in complex VQA and commonsense reasoning tasks, proving its utility beyond Bongard problems. While these standard benchmarks rely on single-instance comparisons --- limiting the use of prototype vectors which we showed in Appendix E provides a better estimate of separability --- the framework still effectively diagnoses the nature of the alignment gap in these complex domains.
>
> We've also moved the conclusions from Appendix A into new "Discussion, limitations and future-work" section right before conclusion, which now more deeply discusses these concepts and the potential future direction also incorporating textual representations.
>
> Finally, we've updated the contributions to be more concise to better reflect the work done.
>
> We hope these additional experiments and discussions sufficiently address your concerns regarding the metric's validity and the work's contributions. Do you have any other concerns or questions?

---

### Official Review · Reviewer_XeiP · 2025-10-29

**Soundness:** 2
**Presentation:** 2
**Contribution:** 3
**Rating:** 4
**Confidence:** 4

**Summary:**

This paper investigates why Visual–Language Models (VLMs) often fail on abstract visual reasoning tasks and whether such failures arise from perception or reasoning deficits. To diagnose this, the authors propose the Linear Separability Ceiling (LSC) framework, which quantifies the performance achievable by a linear probe on a model’s visual embeddings. Analyses across state-of-the-art VLMs reveal a pervasive alignment gap, where models rarely exceed their own LSC. To address this, the study augments next-token prediction with a contrastive objective that enhances the linear structure of visual representations. Fine-tuned models consistently surpass the LSC, achieving human-level accuracy on Bongard-OpenWorld and significant gains on relational reasoning benchmarks, demonstrating that reasoning limitations stem from misalignment rather than intrinsic capacity.

**Strengths:**

1)	The paper introduces the Linear Separability Ceiling framework to disentangle perception and reasoning in VLMs.
2)	Through large-scale analysis, the authors reveal a pervasive alignment gap: most VLMs fail to outperform their own LSC, highlighting a fundamental but previously unmeasured bottleneck in vision–language reasoning.
3)	The approach attains or surpasses human-level accuracy on OpenWorld and narrows the gap on HOI reasoning, demonstrating that the limitation in current VLMs stems from misalignment, not innate capacity.

**Weaknesses:**

1)	While the Linear Separability Ceiling is intuitively defined, the paper lacks a rigorous theoretical justification for why linear separability should represent the upper bound of perceptual quality. A more formal link between LSC and model capacity or information-theoretic limits is missing.
2)	The claim that failures arise from “alignment gaps” rather than perception deficits is mostly correlational. The experiments show association but not causal evidence that reasoning misalignment causes underperformance.
3)	The evaluation focuses mainly on Bongard-style reasoning and a single compositional benchmark. Broader validation on diverse abstract reasoning or real-world multimodal tasks would strengthen the generality of conclusions.
4)	The “nonlinear decision logic” mechanism is described conceptually but not visualized or quantitatively analyzed. Without feature attribution or attention-map evidence, the interpretation remains speculative.
5)	This paper is lengthy and conceptually dense. Core ideas like “alignment gap” and “surpassing the ceiling” could be more precisely illustrated. Figures (e.g., Fig. 2) lack clear axis descriptions, and some tables overflow with statistical detail without clear takeaway messages.
6)	The paper’s structure is somewhat diffuse, with diagnostic and intervention sections interleaving and key concepts repeated across sections. This weakens the logical flow from problem to solution and makes the main argument harder to follow.

**Questions:**

Please refer to the weak points.

---

> ### Author Response · Authors · 2025-11-17
>
> We thank the reviewer for taking the time and going through the paper and suggesting ways of improvements.
>
> Here are our clarifications on the weak points:
> 1. We clarify that the LSC is not an upper bound, but rather a baseline for accessibility. Since the vision encoders (e.g., CLIP, SigLIP) are pre-trained with a contrastive objective that relies on the dot product (a linear operation) between mean pooled embeddings, their features are explicitly optimized to be linearly separable.
> 2. We argue that our baseline results provide more than correlational evidence. The LSC serves as an existence proof: because the linear probe can extract the correct label from the visual embeddings, the information is provably present. Consequently, the model's failure to generate the correct answer is causally linked to the processing (alignment) of those representations, not the representations themselves. For performance beyond that, two approaches exist: improving perception or applying non-linear decision logic.
> 3. We applied our framework to five VQA benchmarks: HOI, GQA, POPE, A-OKVQA, and ScienceQA, detailed in Appendix A (on page 15). Here, the alignment gap was less prominant. So while the framework can be adapted to other tasks, the auxilary contrastive function didn't result in superior performance. There are a lot of variables, however, which warrant deeper investigation.
> 4. Such (non-linear) reasoning pathways can be seen in the last appendix of the paper (Appendix O page 33), where the strong vertical stripes appear from images - salient features which the model uses. Salient feature means the model is capable of feature selection through softmax in attention module, which is a non-linear operation. Postfix tuning is also a good example. When the model was initially unable to reach the right conclusion, adding a learned prompt at the end of the sequence, where the visual representations remained exactly the same as in baseline prompt - the model started performing better. Meaning that the learned input sequence after images was the cause for improved results.
> 5. We’ve addressed these issues in the updated paper. Core concepts are visualized on Figure 2 on page 4. Figures now contain clear axis descriptions and statistical detail was removed from the main body of the paper as it is available in the appendixes of the paper.
> 6. We’ve also improved the narrative flow of the paper: we describe the methodology and diagnostic metrics we use; we present the finding; which we then analyze to find 2 reasoning pathways; we find a dependency between the two; we determine which part of the model is responsible for achieving performance gains; then we show adding an auxiliary contrastive loss connects the two reasoning pathways; we compare the results of two objectives, with remaining flow remaining the same. These changes are on pages from 4 to 8 (inclusive).

---

> ### Author Response · Authors · 2025-11-26
> **Follow-up**
>
> We have further updated the manuscript with additional analyses to further address your concerns regarding theoretical grounding and generality.
>
> To address the need for rigorous justification of the LSC, we added manifold structure visualizations (Isomap, Appendix B). These explicitly show the emergence of 1D linear "rays," providing geometric evidence that such structures are already present in baseline model and our method further structurally refines these.
>
> Broader Validation (Appendix A): We updated the LSC on VQA section to establish LSC baselines for POPE, A-OKVQA, and ScienceQA. This demonstrates that the LSC diagnostic framework is generalizable and effective across diverse reasoning benchmarks beyond Bongard tasks.
>
> We've also added an "Discussion, limitations and future-work" section right before conclusion which more deeply discusses these concerns, so you'll get the full picture from there.
>
> Finally, we've updated the contributions to be more concise to better reflect the work done.
>
> Do these additions, combined with the previous additions, sufficiently clarify the validity and scope of our claims? Do you have any other questions or concerns?

---

### Official Review · Reviewer_jWGQ · 2025-10-31

**Soundness:** 2
**Presentation:** 3
**Contribution:** 3
**Rating:** 6
**Confidence:** 3

**Summary:**

This paper introduces the LSC to diagnose whether VLMs fail on abstract reasoning tasks due to perception or reasoning. The authors identify an alignment gap where reasoning fails to surpass the model’s own representational limit. They further proposed a contrastive fine-tuning objective that can close this gap on Bongard-style reasoning tasks.

Overall, the paper is thoughtfully motivated but overinterprets linear-probe diagnostics as mechanistic evidence for reasoning non-linearity. The statistical evidence is somewhat weak, and the causal link between representational geometry and reasoning behavior remains speculative. I will consider moving up the score, depending on how the authors clarify statistical robustness and theoretical grounding during discussion.

**Strengths:**

1. The paper is well motivated and identifies an important problem: diagnosing perceptual–reasoning misalignment in VLMs.
2. The proposed LSC is a clear, interpretable metric that operationalizes representational quality in a meaningful way.
3. The paper is well written and organized, with 2 dataset on 8 models coupled with various promptings.
4. The paper introduced a contrastive fine-tuning objective that simultaneously improves generative accuracy and final-layer separability.

**Weaknesses:**

1. The paper defines “non-linearity” as “cases where linear probes fail.”  The non-linearity can be an artifact of your measurement of cosine similarity of euclidean averaged embeddings, not a measured representational property. The claim would be stronger with direct evidence of curvature or manifold structure.
2. Important evaluation details are underspecified—for instance, how generative accuracy is computed relative to the probe-based classification accuracy.
3. Some of the results (e.g., Section 7.2) appear cherry-picked without consistent statistical treatment. A group-level comparison across models or prompt conditions would increase confidence in the claims.
4. The causal interpretation—that contrastive fine-tuning resolves misalignment—remains speculative. The improvements may simply reflect more linearly organized feature geometry rather than deeper mechanistic reasoning.
5. The paper notes catastrophic forgetting as a limitation but does not fully analyze why L_{combined} causes this — an area that could benefit from ablation or regularization experiments.

**Questions:**

1. How sensitive are your results to pooling strategy (mean pooling vs. attention pooling vs. CLS token)?
2. How stable is the LSC metric across different random seeds or mini-batch samplings?
3. How exactly is the statistical comparison between generative accuracy and LSC performed in Fig. 2—are these paired comparisons over test trials or aggregated accuracies? For models that pass LSC, the advantage of generative accuracy is low.
4. Could this framework be applied to tasks requiring fine-grained perceptual reasoning (e.g., gaze direction or social interaction), rather than Bongard tasks that suit better for linear separation?

---

> ### Author Response · Authors · 2025-11-17
>
> We thank the reviewer for their thoughtful feedback and the opportunity to strengthen our work. We have updated the manuscript with new experiments and visualizations to address your concerns.
>
> Clarification on the weaknesses:
> 1. We’ve added an image of PCA (Figure 4 on page 7) which displays the embedding space before and after contrastive loss was applied. Before, all the image representations were overlapping and were not well linearly separable for the task at hand.
> 2. Generative performance is achieved through prompting, Appendix B (on page 18) has the specific prompts and relative probe extraction method is now visualized in Appendix C (on page 19).
> 3. We pinpointed the location for intervention on Phi only. Methods we found to work were then repeated on other models as well.
> 4. We found that generative predictions were pair-wise correlated with linear probe results, the accuracies converged, and attention maps (last appendix) visualize how models’ behavior changed. A property of a linearly separable representation is that readouts are computationally simple, so if there is a shortcut in models’ representations during training that predicts the answer perfectly, the models will latch onto that first.
> 5. The models were trained with an aggressive contrastive signal which restructured representations, the model ran into image-to-text alignment gap as the old reasoning pathways didn’t work on the new representations. Section 7 on page 9 restates that, and Appendix A does a deeper dive into these mechanics on VQA datasets.
>
>
> Response for the questions:
> 1. Attention pooling introduces non-linearity as softmax is a non-linear operation, and most VLMs don’t have a special token which summarizes the image, so pooling representations ‘the manual way’ is the most reasonable way forward. And here, mean pooling has been used for this in the past since it preserves angular properties, so we did the same.
> 2. The LSC probe doesn’t have any parameters, so it’s deterministic and gives the same result (floating point errors here and there).
> 3. We use a chi-squared test based on the paired, trial-by-trial correctness. Updated the article to specify that.
> 4. LSC can work with single image/text representations, with prototype vectors to compare against being more exact, so yes. We explored its usage in VQA-type problems in Appendix A where we found no alignment gap most of these tasks. For a better estimate, a prototype vector could give a more exact estimate as we show in Appendix D on page 20.

---

> ### Author Response · Authors · 2025-11-26
> **Follow up**
>
> We have included the manifold structure visualizations (Appendix B) to address your concern regarding potential measurement artifacts, and updated the LSC on VQA (Appendix A) to give depth to the image-to-text alignment issue. Furthermore, we've added a sensitivity analysis (Appendix N) to the auxillary contrastive objective to better address questions regarding training stability and regularization.
>
> We've also added an "Discussion, limitations and future-work" section right before conclusion which more deeply discusses these concerns, so you'll get the full picture from there.
>
> Finally, we've updated the contributions to be more concise to better reflect the work done.
>
> Do these additions sufficiently clarify the theoretical grounding and validity of our claims? Do you have any other questions or concerns?

---

### Author Response · Authors · 2025-12-01
**TL;DR for AC - how we addressed the main reviewer concerns**

Methodology
---

Non-linearity can be an artifact of your specific measurement. (jWGQ)
- PCA (Figure 4 on page 7) visually confirms that baseline representations are geometrically entangled in the dominant variance dimensions (resulting in a low final layer LSC score).

How sensitive are your results to pooling strategy (mean pooling vs. attention pooling vs. CLS token)? (jWGQ)
- attention pooling could find the variance dimension where the representations are better linearly separable, but the entire process involves non-linear softmax operations. So, mean pooling measures the intrinsic linear separability.

Why linear separability should represent the upper bound of perceptual quality? (XeiP)
- Since the vision encoders (e.g., CLIP, SigLIP) are pre-trained with a contrastive objective that relies on the dot product (a linear operation) between mean pooled embeddings, their features are explicitly optimized to be linearly separable.

It's possible that representations hold complex non-linear structures. (3cSn)
- While possible and we fully acknowledge that, we're using LSC as a baseline and not as an upper bound.

Statistical robustness and cherry-picking. (jWGQ)
- We use Chi-squared on paired, trial-by-trial correctness, and the main claims made are all found to be consistent on the 3 selected models.

Findings
---

The claim that failures arise from “alignment gaps” rather than perception deficits is mostly correlational. (XeiP)
- The LSC serves as a proof of existence. Because the linear probe can extract the correct label from the visual embeddings, the information is provably present. Consequently, the model's failure to generate the correct answer is causally linked to the processing (alignment) of those representations, not the representations themselves.

The experiments show association but not causal evidence that reasoning misalignment causes underperformance. (XeiP)
- For performance beyond baseline, two approaches exist: improving perception or applying non-linear decision logic. Non-linear decision logic is proven by postfix tuning where visual representations remain exactly as is, but the reasoning pathways after visual tokens are altered to achieve performance superior to LSC. For representation refinement (improving perception), a contrastive objective was used.

Improvements (from representation refinement) may simply reflect more linearly organized feature geometry rather than deeper mechanistic reasoning. (jWGQ)
- We found that generative predictions were pair-wise correlated with linear probe results, the accuracies converged, and attention maps (last appendix) visualize how models’ behavior changed. Furthermore, a property of a linearly separable representation is that readouts are computationally simple, so if there is a shortcut in models’ representations during training that predicts the answer perfectly, the models will latch onto that first. Finally, we've added Isomap visualizations of the manifold structure across differing neighbor sizes (Appendix B), visualizing the "one-dimensionally linear" representations, where the contrastive objective promotes globally more consistent concepts.

Part of the observed performance gain could be attributed to alignment with the evaluation metric itself. (jDo8)
- This geometric evidence by Isomap shows a deeper level of understanding.

Limitations & Scope
---

The experimental validation is limited to binary image-to-text retrieval variants. (jDo8)
- With the paper being long as it is, we narrowed down the experimental results discussed and respective claims made in the main body of the manuscript to explicitly take that into account.

Additional experiments on independent tasks not directly linked to the contrastive loss would help verify generalization. (jDo8)
- Almost all SoTA VLMs use contrastively trained vision encoders.

The paper notes catastrophic forgetting as a limitation but does not fully analyze why L_{combined} causes this. (jWGQ)
- Contrastive pressure changes the computational pathway the model uses to reach an answer. It does so by altering the visual representations, and a new pathway emerges utilizing these representations. The old pathways which leveraged the old representations no longer work.

Broader validation on diverse abstract reasoning or real-world multimodal tasks would strengthen the generality of conclusions. (XeiP)
- Appendix A evaluates LSC, baseline, and HOI-trained Phi models on 4 multiple-choice VQA tasks.

A sensitivity study or learning dynamics analysis would strengthen the claims. (jDo8)
- Appendix N includes the sensitivity study, which further demonstrates this image-to-text misalignment.

Can the fine-tuning approach be successfully applied beyond Bongard problems?
- Appendix A showcases that the framework can be applied for image-to-text pairs, and we leave alignment fine-tuning on these tasks as future work.

Full answers to all concerns are in their respective threads and manuscript itself.

---

### Meta-Review · Area_Chair_FQMC · 2025-12-27

**Summary:**

This paper received 6,4,4,4. Initial concerns include how the paper defines non-linearity and how it is measured, some missing evaluation details, cherry-picked results, speculative interpretation that contrastive fine-tuning resolves misalignment, lacking a rigorous theoretical justification of why linear separability should represent the upper bound of perceptual quality, limited evaluation, core observation not being new, and clarity.

**Reviewer Concerns:**

Most of the initial concerns were addressed by the rebuttal, however, remaining concerns include lacking rigorous theoretical justification, and generalization results beyond Bongard to e.g., VQA, commonsense tasks. The latter was a common concern/question from multiple reviewers. The new VQA results in the rebuttal do not appear to adequately support the main claims, as the evidence seems insufficient to draw strong conclusions about generalization. The paper would be significantly improved with a deeper analysis on those tasks, rather than as an exploratory evaluation as currently provided during the rebuttal phase.  The ACs would like to encourage the authors to improve their paper by addressing these concerns and resubmit to a future conference.

**Reviewer Scores:**

It is very likely that all of the reviewers would have maintained their scores, given the remaining concerns.  The 6 reviewer may not have increased their score since the rebuttal does not appear to significantly strengthen the paper and the reviewer's confidence was not high (3).

---

### Decision · Program_Chairs · 2026-01-26

Reject